


# Analysis of multiple new-particle growth pathways observed at the US DOE Southern Great Plains field site

Anna L. Hodshire[1], Michael J. Lawler[2, a], Jun Zhao[3, b], John Ortega[2], Coty Jen[3, c], Taina Yli-Juuti[4], Jared F. Brewer[1], Jack K. Kodros[1], Kelley C. Barsanti[5, d], Dave R. Hanson[6], Peter H. McMurry[3], James N. Smith[7], Jeff R. Pierce[1]

[1]Department of Atmospheric Science, Colorado State University, Fort Collins, 80523, USA
[2]Atmospheric Chemistry Observations & Modeling, National Center for Atmospheric Research, Boulder, 80305, USA
[3]Department of Mechanical Engineering, University of Minnesota-Twin Cities, Minneapolis, 55455, USA
[4]Department of Applied Physics, University of Eastern Finland, Kupio, FI-70211, Finland
[5]Department of Civil and Environmental Engineering, Portland State University, Portland, 97201, USA
[6]Department of Chemistry, Augsburg College, Minneapolis, 55454, USA
[7]Department of Chemistry, University of California, Irvine, 92697-2025, USA
[a]now at Department of Chemistry, University of California, Irvine, 92697-2025, USA
[b]now at School of Atmospheric Sciences, Sun Yat-sen University, Guangzhou, 510275, China
[c]now at Environmental Science, Policy, and Management, University of California, Berkeley, 94720-3114, USA
[d]now at Chemical and Environmental Engineering, University of California, Riverside, 92521, USA

*Correspondence to:* Anna L. Hodshire (hodshire@rams.colostate.edu)

**Abstract.** New-particle formation (NPF) is a significant source of aerosol particles into the atmosphere. However, these particles are initially too small to have climatic importance and must grow, primarily through net uptake of low-volatility species, from diameters ~1 nm to 30-100 nm in order to potentially impact climate. There are currently uncertainties in the physical and chemical processes associated with the growth of these freshly formed particles that lead to uncertainties in aerosol-climate modeling. Four main pathways for new-particle growth have been identified: condensation of sulfuric acid vapor, condensation of organic vapors, uptake of organic acids through acid-base chemistry in the particle phase, and accretion of organic molecules in the particle phase to create a lower-volatility compound that then contributes to the aerosol mass. The relative importance of each pathway is uncertain and is the focus of this work.

The 2013 New Particle Formation Study (NPFS) measurement campaign took place at the DOE Southern Great Plains (SGP) facility in Lamont, Oklahoma, during spring 2013. Measured gas-and particle-phase compositions during these new-particle growth events suggest three distinct growth pathways: (1) April 19 shows growth by organics alone; (2) May 9 shows growth by sulfuric-acid/ammonia; and (3) May 11 shows growth by sulfuric-acid/amines/organics. To supplement the measurements, we used the particle-growth model MABNAG (Model for Acid-Base chemistry in NAnoparticle Growth) to gain further insight into the growth processes on these three days at SGP. MABNAG simulates growth from (1) sulfuric-acid condensation (and subsequent salt formation with ammonia or amines); (2) near-irreversible condensation from non-reactive extremely-low-volatility organic compounds (ELVOCs); and (3) organic-acid condensation and subsequent salt formation with ammonia or amines. MABNAG is able to corroborate the observed differing growth pathways, while also predicting that ELVOCs contribute more to growth than organic salt formation. However, most MABNAG model simulations tend to





underpredict the observed growth rates; this underprediction may come from neglecting the contributions to growth from semi-to-low-volatility species or accretion reactions. Our results suggest that in addition to sulfuric acid, ELVOCs are also very important for growth in this rural setting. We discuss the limitations of our study that arise from not accounting for semi- and low-volatility organics, as well as nitrogen-containing species beyond ammonia

and amines in the model. Quantitatively understanding the overall budget, evolution, and thermodynamic properties of lower-volatility organics in the atmosphere will be essential for improving global aerosol models.

## 1. Introduction

10        Atmospheric aerosols can affect climate directly, through the absorption and scattering of solar radiation (Rosenfeld et al., 2008; Clement et al. 2009), and indirectly, by modifying cloud properties (Charlson et al., 1992). Both of these effects depend on aerosol particle size, with particles with diameters larger than 50-100 nm dominating the effects. Larger particles scatter and absorb radiation more efficiently than smaller particles (Seinfeld and Pandis, 2006), and particles with diameters larger than 50-100 nm have the potential to act as cloud

condensation nuclei (CCN; a full list of all abbreviations used in the paper is listed in Appendix A) (e.g., Seinfeld and Pandis, 2006). CCN number and activity are instrumental in determining cloud properties, including precipitation and albedo (Rosenfeld et al., 2008; Forster et al., 2007). The predictions of these aerosol impacts on climate remain amongst the largest uncertainties in climate models (Boucher et al., 2013). Thus, in order to better constrain the climate effects of aerosols, atmospheric particle size distributions must be accurately modeled.

20        The majority of atmospheric aerosols originate from photochemically driven new-particle formation (NPF) (e.g., Spracklen et al., 2008; Pierce et al., 2009). NPF is regularly observed to occur throughout the troposphere (e.g. Kulmala et al., 2004; Kuang et al., 2010). We distinguish between nucleation and NPF as follows: Nucleation is the formation of stable particles ~1 nm in diameter from gas-phase sulfuric-acid molecules and stabilizing vapors that could include water, ammonia, amines, diamines, and oxidized organic molecules (e.g. Kirkby et al., 2011; Chen et

al., 2012; Almeida et al., 2013; Riccobono et al., 2014; Jen et al., 2016). NPF, however, includes the growth of these stable nuclei to sizes frequently observed in the atmosphere (larger than 3-10 nm). In order to grow to climate-relevant sizes, new particles must grow through uptake of vapors while avoiding being lost to coagulation by larger particles. This competition between growth and coagulational scavenging depends primarily on initial and final particle size, growth rate, and the concentration of pre-existing aerosols (Kerminen et al., 2004; Pierce et al., 2007;

Kuang et al., 2010; Westervelt et al., 2013; Westervelt et al., 2014). Large impacts of NPF on CCN are most favorable under conditions of fast particle growth rates and low pre-existing aerosol concentrations (small coagulation sinks). Thus, it is important to understand both particle growth and the time-evolving particle size distributions in order to model the resulting CCN concentrations from new-particle events accurately. In this work, we focus upon the growth of particles from these NPF events.

35        Particle growth from NPF events is chemically complex and poorly understood. Irreversible condensation of sulfuric acid vapor (produced through gas-phase oxidation of $SO_2$ by the hydroxyl radical) is known to be a major contributor to growth. The effective equilibrium vapor pressure of sulfuric acid in the presence of tropospheric water vapor is negligible compared to ambient sulfuric acid concentrations (Marti et al., 1997), and sulfuric acid readily




condenses to the smallest stable particles. However, observed particle growth often exceeds that which can be explained by the condensation of sulfuric acid alone (Weber et al., 1997; Stoltzenburg et al., 2005; Riipinen et al., 2007; Iida et al., 2008; Kuang et al., 2010; Smith et al. 2010; Pierce et al., 2012). These and other studies have shown that the uptake of low-volatility organic vapors is also important and even explains the majority of growth for

5       some regions (e.g., Smith et al., 2008; Kuang et al., 2009; Riipinen et al., 2011; Bzdek et al., 2014; Xu et al., 2015). Growth by organics may involve a large number of species and multiple growth pathways (Riipinen et al., 2012). We propose that particle growth rate can be modelled as the sum of the following processes: irreversible condensation of sulfuric acid ($GR_{H2SO4}$), reversible or nearly irreversible condensation of semivolatile or low-volatility organic compounds ($GR_{org\ cond}$), uptake of organic acids through acid-base chemistry in the particle-phase

($GR_{acid-base}$), and growth from the accretion of two or more organic molecules in the particle phase to form a lower-volatility compound that can then contribute to aerosol mass ($GR_{accret}$):

$$GR = GR_{H2SO4} + GR_{org\ cond} + GR_{acid-base} + GR_{accret} \qquad (1)$$

The contribution of atmospheric vapors to observed growth rates through condensation of these organic vapors (without reactions in the particle phase) depends heavily upon the volatility of the organics in the gas phase. It is estimated that the equilibrium vapor pressure required for near-irreversible condensation of vapors onto nanoparticles (defined here to be aerosol particles with an ambient diameter less than 50 nm) must be around $10^{-7}$ Pa ($\sim10^{-12}$ atm) or less, corresponding to a saturation concentration of $10^{-4}$-$10^{-3}$ µg m$^{-3}$ (Donahue et al., 2011; Pierce et

al., 2011).

        The presence of essentially non-volatile organic vapors in the atmosphere, referred to here as extremely low-volatility organic compounds (ELVOCs), defined to have saturation concentrations of around $10^{-4}$ µg m$^{-3}$ or less (Murphy et al., 2014), have been detected in both laboratory and ambient measurements (Ehn et al., 2012; Zhao et al., 2013, Jokinen et al., 2015). Ehn et al. (2014) proposed a possible chemical pathway in which large atmospheric

organic parent molecules (e.g. terpenes) undergo initial oxidation to form peroxy radicals followed by rapid autoxidation (self reaction), creating highly oxygenated, yet still large (e.g., 10 carbons) molecules. This pathway has since been confirmed by Jokinen et al. (2014) and Rissanen et al. (2014). Jokinen et al. (2015) determined ELVOC yields from five major biological volatile organic compound (BVOC) species from both ozonolysis and OH oxidation, including isoprene and 4 monoterpenes (limonene, alpha-pinene, myrcene, and beta-pinene). The ELVOC

yield for isoprene from the two oxidation pathways is low (0.01% and 0.03%, respectively); however, since isoprene emissions are the highest among all non-methane BVOCs (Guenther et al., 2004), these pathways could contribute an appreciable amount of ELVOCs in high isoprene-emitting regions (e.g., Yu et al., 2014). The monoterpenes have much higher ELVOC yields, ranging from 0.12% to 5.3%, depending on both the monoterpene structure and oxidation pathway. Subsequent global aerosol simulations have indicated that the ELVOCs produced from the

observed monoterpene yields increased NPF and growth globally, which in turn increased total number concentrations across most of the continental regions and moderately increased the number of CCN (Jokinen et al., 2015).



Ammonia can form inorganic salts in atmospheric particles with sulfuric acid and nitric acid (Jaeschke et al., 1998; Seinfeld and Pandis, 1998); these reactions shift the equilibrium of ammonia (the partitioning species) to the particle phase, as the inorganic salts are lower in volatility than their individual constituents (Pankow, 2003; Pinder et al. 2007). Amines (nitrogen-containing bases with at least one carbon) and organic acids also are observed

in growing new atmospheric particles (e.g., Mäkelä et al, 2001; Smith et al., 2008; Smith et al., 2010; Wang et al., 2010; Tao et al., 2015). Since the vapor pressures of these compounds are higher than is favorable for contributing to new-particle growth by non-reactive condensation alone, the formation of organic salts (formed from organic acids reacting with either amines or ammonia) has been suggested as a potential mechanism for reducing the volatility of these compounds (Barsanti et al., 2009). The presence of these organic-acid and base species in the

particle phase depends on the thermodynamic properties of these species (volatility and pKa) (Barsanti et al., 2009) as well as the amount of sulfuric acid, which will preferentially react with bases.

Accretion products are formed from a large variety of reactions, through which organic molecules may contribute to particle mass by reactions between organic molecules that reduce the volatility of the parent molecules (Barsanti and Pankow, 2004; Pun and Seigneur, 2007). Assessing the tendency of atmospheric molecules to undergo

accretion reactions via thermodynamic considerations showed that glyoxal and methylglyoxal and acetic, malic, maleic, pinic, and likely other similar mono- and di-carboxylic acids have the thermodynamic potential to contribute significantly to particle growth under the right kinetic conditions (Barsanti and Pankow, 2004; 2005; 2006). Matsunaga et al. (2005) found that small multifunctional compounds (e.g. methylglyoxal) in the ambient atmosphere had a much higher particle-phase affinity than predicted by their Henry's law constants; they proposed

oligomerization as a possible pathway. Several laboratory studies have confirmed the presence of accretion products in secondary organic aerosols (SOA) formed from a variety of precursor species (Limbeck et al., 2003; Tolocka et al., 2004; Heaton et al., 2007; Wang et al., 2010). On a mass basis, polymers and oligomers have been found to account for up to 50% of the SOA formed from ozonolysis (Gao et al., 2004; Kalberer et al., 2004; Hall and Johnston et al., 2011). Wang et al. (2010) directly observed oligomers from glyoxal reactions in growing particles

from 4-20 nm in diameter, indicating that accretion products have the potential to contribute to new-particle growth. While there are studies showing that accretion could be an important process for particle growth, there are still many uncertainties associated with it.

Despite the growing body of evidence for multiple growth pathways for new-particle growth, current global and regional model studies of aerosol impacts focus on growth through the condensation of vapors only, generally

sulfuric acid and lumped organics (e.g. Yu et al., 2011; D'Andrea et al., 2013; Jokinen et al., 2015; Scott et al., 2015). Often, global and regional models with online aerosol microphysics have made simplified assumptions about SOA yields and the size-dependent uptake of organic vapors to particles. Many microphysics models assume fixed SOA yields (e.g., Pierce et al, 2009; Spracklen et al., 2010; Spracklen et al., 2011; Westervelt et al., 2013), as size- and volatility-resolved vapor condensation/evaporation is a computationally burdensome system; others explicitly

include volatility-dependent yields (e.g., Zaveri et al., 2008; Yu et al., 2011). The fixed-yield models either treat SOA as ideally semi-volatile, with the assumption that organic vapors reach instantaneous equilibrium with the aerosol and condense proportionally to the pre-existing particle mass distribution, or the models assume that the





SOA is effectively non-volatile and condenses proportionally to the pre-existing Fuchs-corrected surface area (Pierce et al., 2011; Riipinen et al., 2011; Zhang et al., 2012a).

Generally, regional and global models do not account explicitly for the possible particle-phase reactions (organic acid-base chemistry and oligomerization) with some exceptions (e.g. Carlton et al 2010). To our knowledge, no regional or global modelling study has investigated the role of these particle-phase reactions on new-particle growth. The studies discussed above are simply attempting to account for all growth via traditional non-reactive gas-phase condensation. On the other hand, there are several process-based box models that implicitly or explicitly simulate particle-phase processes in addition to condensation and non-reactive partitioning, including the oligomer formation framework of Pun and Seigneur (2007) and Ervens et al. (2010); the kinetic modelling framework of Pöschl et al. (2007), extended by Shiraiwa and co-workers to build multi-layer kinetic models of gas-aerosol interactions (Shiraiwa et al., 2009; 2010; 2012); and the Model for Acid-Base chemistry in NAnoparticle Growth (MABNAG; Yli-Juuti et al. (2013)), a single-particle growth model that simulates particle-phase acid-base reactions as well as condensation/evaporation. These detailed, process-based aerosol models may be used to determine the relative contributions of the various potential growth pathways ($GR_{H2SO4}$, $GR_{org\ cond}$, $GR_{acid-base}$, $GR_{accret}$) but to our knowledge have not been used extensively in conjunction with detailed measurements of growth events. Ultimately, well-tested and measurement-informed process-based models should be used in the future to create next-generation particle-growth schemes for more realistic global and regional aerosol models.

In this study, we seek to understand the species and mechanisms that drove the growth of new particles observed during the Southern Great Plains (SGP) New Particle Formation Study (NPFS) in April-May 2013 in Oklahoma, USA. We attempt to find closure in particle growth rates and particle composition between a state-of-the-art process-based growth model (MABNAG) and detailed measurements of particle growth, particle composition, and gas-phase species. We consider $GR_{H2SO4}$, $GR_{org\ cond}$, and $GR_{acid-base}$. We do not consider $GR_{accret}$ as we do not have sufficient measurements to constrain these rates. Through this closure process, we provide estimates of the dominant species and mechanisms for three specific growth events observed during the study. Section 2 provides an overview of our measurement and modelling methods. Section 3 closely examines three NPF events observed during the NPFS at SGP and compares these events to modelling results using MABNAG. Conclusions and future work are discussed in Section 4.

## 2. Methods

The Southern Great Plains (SGP) New Particle Formation Study (NPFS) took place from April 13 to May 24, 2013 (http://www.arm.gov/campaigns/sgp2013npfs). The primary objectives of the campaign were to study the formation and evolution of aerosols and the impacts of the newly formed particles on cloud processes. The majority of the measurements (and all of those used in this work) took place at the US Department of Energy (DOE) Atmospheric Radiation Measurement (ARM) SGP Central Facility in the Guest Instrument Facility. The site is representative of the large Great Plains region, with agricultural activities, such as cattle and pig husbandry, as well as oil and gas extraction. To our knowledge, the nucleation and growth in the Great Plains region has not been studied in detail. For more information on the site and campaign, visit the DOE and campaign report websites



(http://www.arm.gov/sites/sgp and http://www.arm.gov/campaigns/sgp2013npfs).  Thirteen new-particle formation events were observed during the NPFS. In this paper, we focus on three new-particle formation events that occurred on April 19, May 9, and May 11; these were the days where NPF was observed and all the available equipment was operating properly.  Figure 1 shows the observed size distributions and derived back trajectories from the HYbrid

Single-Particle Lagrangian Integrated Trajectory (HYSPLIT) model (Draxler and Rolph, 2012; Rolph, 2012) for these three days.  These data will be described in detail later.

### 2.1 Measurements

During the 6-week campaign, 13 new-particle formation events were observed at Lamont by a battery of three Scanning Mobility Particle Sizers (SMPS) operated in parallel. They included the DEG SMPS (a TSI 3085 Nano DMA operated with a laboratory prototype laminar flow diethylene glycol condensation particle counter detector; Jiang et al.(2011); 1.9-13.6 nm mobility diameter), a Nano SMPS (a TSI 3085 Nano DMA operated with a TSI 3025A laminar flow ultrafine butanol CPC detector; 2.8-47 nm mobility diameter), and a conventional SMPS (a

home-built long column DMA with dimensions similar to the TSI 3071 with a TSI 3760 CPC detector; 23-528 nm mobility diameter).  For all systems, filtered ambient air was used for the DMA sheath air, without adjusting the water vapor partial pressure.

Nanoparticle composition data were collected using the Thermal Decomposition Chemical Ionization Mass Spectrometer (TDCIMS) (Voisin et al., 2003; Smith et al., 2004). For the observations reported here, we used the

recently developed time-of-flight mass spectrometer version of the instrument (TOF-TDCIMS) (Lawler et al., 2014). The TDCIMS measures the molecular composition of size-selected atmospheric nanoparticles in near-real-time. It performs this measurement by first charging and size-selecting nanoparticles using unipolar chargers and differential mobility analyzers, respectively. Charged, size-selected particles are collected by electrostatic precipitation onto a platinum filament for approximately 30 min. Following this, the filament is moved into the ion

source of a chemical ionization mass spectrometer and undergoes a current ramp to reach an estimated maximum temperature of 600 °C. This heating thermally desorbs and/or decomposes the sample to produce gas phase analyses. Two different chemical ionization reagents are used to detect the chemical species desorbed from the sample: $H_3O^+(H_2O)_n$ (n=0-3), hereafter referred to as "positive ion chemistry", detects base compounds such as ammonia and amines as well as carbonyl-containing compounds and some alcohols; $O_2^-(H_3O)_n$ (n=0-3), hereafter referred to

as "negative ion chemistry", detects organic and inorganic compounds with acid groups, as well as other oxygenated compounds with high electron affinities. During the campaign, the instrument cycled roughly hourly between positive and negative ion chemistry. We classify the detected ions into the following categories: ammonia, amine/amide, organics with sulfur, organics with nitrogen, organics without sulfur or nitrogen, sulfate, and nitrates that are either oxidized (no carbons) or inorganic (see Figs. 2-4, panels (c)-(d)).  At the present time, we have not

identified marker compounds for the condensation of ELVOCs; however, a prior laboratory study has shown that the detection of organic acids in nanoparticles correlates with the early growth of nanoparticles from the oxidation of a-pinene (Winkler et al., 2012). We are also unable to distinguish between the oxidized nitrates and the inorganic nitrates; thus we have grouped these ions together (the nitrate (ox/inorg) category in Figs. 2-4, panels (c)-(d)).



Ambient gas-phase sulfuric acid, malonic acid, and oxalic acid were measured with the Cluster CIMS using nitrate core ion (present primarily as dimer, $HNO_3 \bullet NO_3^-$) as the chemical ionization reagent ion (Zhao et al., 2010). Sulfuric acid, malonic acid, and oxalic acid were detected at m/z 160, 166, and 152 respectively (the molecules clustered with a nitrate ion). The Cluster CIMS measures those acids with unit mass resolution. While the detection

of sulfuric acid in the CIMS has been quantified and calibrated, the detection of oxalic acid and, to a much lesser extent, malonic acid may not be as efficient as sulfuric acid due to gas-phase proton affinities of the organic acids compared to that of nitric acid. A calibration comparison with a different Cluster CIMS using acetate ($CH_3CO_2H \bullet CH_3CO_2^-$) as the reagent ion (Jen et al., 2015) showed up to two orders of magnitude higher inferred oxalic acid concentration and approximately similar malonic concentrations as the nitrate Cluster CIMS. Therefore,

the estimated systematic uncertainty in the oxalic acid measured via nitrate chemical ionization is approximately a factor 100 lower. We explore the sensitivity of the model to these organic-acid uncertainties in this paper.

Ambient gas-phase amines and ammonia concentrations were measured using the Ambient pressure Proton transfer Mass Spectrometer (AmPMS) (Hanson et al., 2011; Freshour et al., 2014), a quadrupole instrument (unit mass resolution) with high sensitivities for ammonia and amines. Signals at the protonated parent masses for

methylamine, dimethylamine, and trimethylamine (C1-C3 amines) were assigned with confidence; also detected was a suite of larger alkylamines with four to seven carbons (C4-C7). Less is known about the speciation of these larger amines, as ambient measurements of amines larger than C3 are not often made (e.g. Ge et al., 2011). Contribution of amides to the signals at the masses of the larger amines may also be significant; as such, no structure information was assigned to the C4-C7 amines, as many isomers are possible.

A Proton Transfer Reaction Mass Spectrometer (PTR-MS) based on the design of Hanson et al. (2011) was operated unattended during the campaign and was set to measure a suite of volatile organic compounds (VOCs), including isoprene and monoterpenes. However, only one calibration was done for the PTR-MS on May 18, 35 days into the campaign, and during processing, unexplainable spikes were seen in the data at irregular intervals. Further, monoterpene mixing ratios were nearly always unreasonably high (often ranging between 10-100 ppbv). For

comparison, a field site in Manitou, Colorado, comprised of a ponderosa pine stand, had maximum monoterpene mixing ratios of 1-2 ppbv during the mid-summer (Ortega et al., 2015), and we expect the monoterpene emissions near the SGP (with few trees) site in April and May to be lower than the forested Manitou site in summer. We thus lack confidence overall in the VOC data obtained by the PTR-MS, so we use an alternative method for estimating monoterpene concentrations, which is described below.

**2.2 ELVOC estimate**

Rather than using the PTR-MS for VOC data, which suffered from calibration issues, we estimate monoterpene emissions and concentrations using the Model of Emissions of Gases and Aerosols from Nature

version 2.1 (MEGAN2.1) (Guenther et al, 2006; Guenther et al, 2012; Sindelarova et al., 2014) in the Goddard Earth Observing System chemical-transport model (GEOS-Chem; http://geos-chem.org). We ran MEGAN2.1 in GEOS-Chem at a 2x2.5 degree resolution to estimate monoterpene emissions rates (monoterpenes are not tracked as prognostic species in these GEOS-Chem simulations). The specific monoterpenes estimated are alpha-pinene, beta-





pinene, limonene, sabinene, myrcene, 3-carene, ocimene, and the lumped sum of other monoterpenes (see Guenther et al., 2012 for a complete list). These GEOS-Chem simulations use GEOS-FP meteorological fields generated by the Goddard Modeling and Assimilation Office (GMAO, http://gmao.gsfc.nasa.gov/) and include biogenic emission-factor updates to MEGAN2.1 based on Guenther et al. (2012) and Sindelarova et al. (2014). We estimate pseudo-

steady-state monoterpene concentrations by assuming that the emitted monoterpenes are well mixed up to the boundary-layer (BL) height measured at SGP, and that emissions are balanced by chemical loss by ozonolysis. (The BL height measurements were obtained by the ARM value-added product radiosonde (PBLHTSONDE) at the SGP Central Facility.) For ozonolysis, we used a rate constant, $k$, of $8.1 \cdot 10^{-17}$ cm$^3$ molecule$^{-1}$ s$^{-1}$ for all monoterpenes, from IUPAC (http://www.iupac.org). For the ozone concentrations, we used hourly ozone monitor measurements

from the closest EPA monitoring site, at Dewey, OK, which is 120 miles SW of the SGP site. The uncertainty in ozone concentration due to the distance between measurements is a source of potential error in our monoterpene concentration calculation; however, since we expect ozone concentrations to be relatively homogeneous regionally, we expect other errors (such as ELVOC yields), to be more significant sources of ELVOC uncertainty.

We estimate the gas-phase ELVOC from the oxidation of the monoterpene (MT) concentrations obtained

from MEGAN, assuming a pseudo-steady state between its chemical production and loss by irreversible condensation and neglecting dry deposition as the condensationsink timescales are faster than the dry-deposition timescales (Pierce and Adams, 2009):

$$[ELVOC] = \frac{0.03\, k\, [O_3]\, [MT]}{CS} \tag{2}$$

where CS is the condensation sink, calculated from the SMPS aerosol size-distribution measurements. We note that the SMPS measurements only go up to ~650 nm mobility diameter, so the condensation sink calculated represents a lower limit on the actual condensation sink. The prefactor, 0.03, is the ELVOC molar yield from the α-pinene + ozone reaction found in Jokinen et al. (2015). Alpha-pinene represents ~30% of the MEGAN-estimated

monoterpenes present at SGP during the campaign, which is the largest fraction by any of our estimated monoterpene species. Thus, we assume the α-pinene yield to be representative of all of the monoterpenes; in reality, some monoterpene species have higher or lower yields. We do not know the ELVOC yield from oxidation processes for all monoterpene species; thus, this estimate of the ELVOC concentration should be taken as one possible outcome of monoterpene oxidation. We also acknowledge that our modelled monoterpene concentrations depend on

the modelled ozone concentrations, whereas we used the measured ozone concentrations to determine the ELVOC yield. The errors in this assumption are likely small compared to our uncertainties in the ELVOC yield.

### 2.3.1 Model description

The Model for Acid-Base chemistry in NAnoparticle Growth (MABNAG) has been developed by Yli-Juuti, et al (2013) to simulate the growth and composition of a single particle resulting from both condensation of low-volatility vapors and acid-base reactions in the particle phase. The version of MABNAG used for this study accepts as inputs the gas-phase concentrations and properties of water, sulfuric acid, a representative organic acid,



ammonia, a representative amine, and a representative non-reactive organic, taken here to be an extremely-low-volatility organic compound (ELVOC). The organic compounds are represented in MABNAG with the chemical properties (e.g. pKa, molecular mass, equilibrium vapor pressure) of one organic acid, one amine, and one ELVOC; thus, we must make assumptions about the properties of the organic acid, amine, and ELVOC inputs that are

representative for the wide ranges of organic-acid, amine, and ELVOC species. MABNAG also requires an initial particle size and composition; for simplicity in this study, the initial particle is formed from 20 molecules of each input species. We assume a particle density of 1.5 g cm$^{-3}$ and a surface tension of 0.03 N m$^{-1}$. A sensitivity case using 0.05 N m$^{-1}$ for the surface tension did not affect our results at the particle diameters where we compare to measurements (above 10 nm).

10          The uptake rates of sulfuric acid, the organic acid, and the ELVOC are calculated as gas-phase-diffusion-limited mass transfer based on their ambient vapor pressures, equilibrium vapor pressures, and gas-phase diffusivities. Water and the bases are assumed to instantly reach equilibrium between the gas and particle phases due to their higher diffusivities and pure-species vapor pressures. Upon uptake, subsequent acid dissociations and base protonations in the particle phase are calculated by the Extended Aerosol Inorganics Model (E-AIM)

(http://www.aim.env.uea.ac.uk/aim/aim.php, Clegg et al., 1992; Clegg and Seinfeld, 2006a, b; Wexler and Clegg, 2002). It is assumed that the ELVOC does not dissociate in the particle phase. This vapor pressure is low enough that uptake of ELVOCs is essentially irreversible, even at the smallest simulated particle sizes. We do not consider any additional particle-phase reactions beyond the acid-base reactions:  this includes possible accretion reactions that could contribute to growth. We have estimated ELVOC concentrations as they have been shown to have a direct

oxidation pathway from monoterpene species to ELVOC species (e.g. Jokinen et al., 2015). On the other hand, the estimation of SVOC concentrations and oligomerization is much less constrained: one must know how the SVOCs are reversibly partitioning to the full aerosol size distribution (as opposed to irreversible condensation to the condensation sink for ELVOCs), and oligomerization rates and the involved SVOC species are highly uncertain. For these reasons, we will not attempt to estimate the SVOC concentration present at SGP and will neglect

oligomerization reactions in this work. SVOCs may also directly contribute to particle growth through condensation, as can low-volatility organic compounds (LVOCs), organics with saturation concentrations of 10$^{-1}$-10$^{-3}$ µg m$^{-3}$ (Murphy et al., 2014). The condensation of SVOCs and LVOCs depends on particle size; the likelihood of irreversible condensation increases with increasing particle size (Pierce et al., 2011). Pierce et al. (2011) estimates that SVOCs and LVOCs can begin contributing to particle growth at diameters as small as ~10 nm and ~3 nm,

respectively, but there are still considerable uncertainties as to the extent in which LVOCs and SVOCs partition to these smaller particle sizes. Thus, omitting LVOCs, SVOCs and resultant condensational growth and/or oligomerization reactions from SVOCs that contribute to growth is a limitation of this study and will be discussed further in the conclusions.

           MABNAG assumes that species that enter the particle are instantaneously and homogeneously mixed into a

liquid particle phase. This ignores potential particle-phase diffusion limitations that can arise from heterogeneous particle phases. SOA has been observed to have solid and semi-solid phases in both the laboratory and the field (Virtanen et al., 2010; Virtanen et al., 2011). Riipinen et al. (2012) estimated the importance of potential diffusion




limitations as a function of size: they argue that diffusion does not limit growth for particles smaller than 20 nm diameter but is potentially important for particles 20-50 nm. However, this remains an uncertainty, and we will address this later.

5 **2.3.2 Model inputs**

Inputs to MABNAG were: the gas-phase concentrations from observations or MEGAN-based modelling (Table 1) and chemical properties (Table 2) of water, sulfuric acid, ammonia, an amine, an organic acid, and a non-reactive organic. Relative humidity (RH) is used as a proxy for the water concentration and was obtained from the 60-m tower data maintained by ARM at the Central Facility. Atmospheric temperature was also obtained from the 60-m tower data. The SGP measurement data described earlier provides the gas-phase concentrations of sulfuric acid; ammonia; a suite of amines; and two organic acids, malonic and oxalic acid. The non-reactive organic input will be our ELVOC concentration estimate from the MEGAN monoterpene emissions. ELVOCs consist of a large range of high-molecular-weight compounds with currently unknown structures (Ehn et al., 2014). We assume that our representative ELVOC is one of the dominant ELVOC monomer peaks seen in the mass spectra measured by Ehn et al. (2014), $C_{10}H_{16}O_9$, molecular weight of 280 g mol$^{-1}$, with the possible structure of three COOH groups, 4 CH groups, 3 CH$_2$ groups, and 3 OH groups. (Chemical structure is required for the UNIFAC activity coefficient calculations in E-AIM in MABNAG.) However, as the vapor pressure of this ELVOC is extremely low (assumed to be 10$^{-9}$ Pa), simulations are generally insensitive to ELVOC chemical structure. No direct measurements have been made for the saturation vapor pressure of ELVOCs; we assume a saturation vapor pressure of $1 \cdot 10^{-9}$ P (corresponding to a saturation concentration (C*) of $1.2 \cdot 10^{-4}$ µg m$^{-3}$ at 283 K). This vapor pressure is low enough that uptake of ELVOCs is essentially irreversible, even at the smallest simulated particle sizes.

MABNAG currently simulates one amine and one organic acid, so we ran a suite of sensitivity cases to assess the range of atmospheric acid and base conditions that could help explain observed particle growth. For the amine input, we tested the chemical properties of two amines with single amino groups: dimethylamine (DMA) or trimethylamine (TMA). We denote these cases as DMA and TMA. The pKas of these species are 10.7 (DMA) to 9.8 (TMA), so amines within this pKa range are represented in our sensitivity studies. We tested the sensitivity to the amine concentration input by using the sum of the light amines only (methylamine, DMA, and TMA only; denoted as L) or the sum of all the amines measured (including the C4-C7 amines but excluding the diamines; denoted as T) as the input. For the L cases, we used the chemical properties of DMA or TMA (denoted DMA_L and TMA_L, respectively). We assumed the larger amines, which made up over 50% of the total amines (by mass), have a lower pKa than the light amines, and therefore use properties similar to that of TMA for the T cases (denoted TMA_T). This prevents the over-estimation of the potential contribution of large amines due to salt formation. The assumption that all larger amines behave similarly with low pKas is likely true for alkylamines with a single amino group but does not apply for diamines. Future studies need to examine how diamines react with acids (e.g. dicarboxylic acids to form nylons) and contribute to nanoparticle growth. Regardless, the range of amine pKas and concentrations examined here illustrate the sensitivity of particles to various parameters. For the organic-acid input, we tested using the chemical properties of oxalic or malonic acid, as these were the organic-acid species measured at SGP. These



cases are denoted as OX or MAL. We acknowledge that there is a large range of organic acids in the atmosphere, and other monocarboxylic and dicarboxylic acids have been measured in ambient particles (e.g. Rogge et al., 1993; Sempere et al., 1994; Khwaja et al. 1995; Kawamura et al., 1996; Limbeck and Puxbaum, 1999). However, aerosol data from urban, rural, and remote regions has shown that malonic acid tends to be among the dominant organic-acid species in the particle phase, with oxalic acid as the dominant organic-acid aerosol species at all measurement locations (e.g. Grosjean, 1978; Kawamura and Ikushima, 1993; Rogge et al., 1993; Sempere et al., 1994; Kawamura et al., 1995; Khwaja et al., 1995; Kawamura et al., 1996; Kawamura and Sakaguchi, 1999; Limbeck and Puxbaum et al., 1999; Kerminen et al., 2000; Narukawa et al., 2002; Mochida et al., 2003; Sempere and Kawamura, 2003). Thus, we estimate that the contribution of organic acids predicted by MABNAG represents a lower bound of the total contribution of organic acids to particle growth but might be a reasonable estimate.

There is uncertainty in the saturation vapor pressures of organic acids. A review of dicarboxylic acids and complex mixtures compiled by Bilde et al. (2015) shows the best fit saturation vapor pressure of the subcooled liquid states of malonic and oxalic acid as functions of temperature (Figures 7 and 8 of the review). As there are variations between different reported measurements at the same temperature, we have selected to use the saturation vapor pressure values for the subcooled liquid states of oxalic and malonic acid obtained from the best-fit functions in Bilde et al. (2015). Additionally, we include a sensitivity case of reducing the saturation vapor pressures by one order of magnitude below the values shown in Table 2. This reduction is within the range of uncertainty in Bilde et al. (2015). We denote simulations using the properties of oxalic acid with the saturation vapor pressure reduced by one order of magnitude as OX_LoVP; we use similar notation for the malonic-acid cases (MAL_LoVP).

We further performed sensitivity studies for the concentration of oxalic acid. Due to the uncertainty in the oxalic-acid detection efficiency from the Cluster CIMS, the real oxalic-acid concentration could be up to 100x the reported concentration (Figures 2-4, panel a). Thus, we ran three sets of concentration input tests: the sum of the reported malonic and oxalic acids (denoted as 1ox), the sum of the reported malonic and 10x the oxalic-acid concentration (denoted as 10ox), and the sum of the reported malonic and 100x the oxalic-acid concentrations (denoted as 100ox). Note that since our simulations include the sum of the oxalic-acid and malonic-acid concentrations, the scaling of the oxalic-acid concentrations implicitly also allows for testing uncertainties in malonic-acid uncertainties, although we believe these to be smaller (Eisele and Tanner, 1993). Thus, we tested three dimensions of sensitivities for the organic-acid inputs: pKa, vapor pressure, and organic-acid concentrations.

In total, there are 36 sensitivity cases for each day (Tables 4-6). We present the case MAL/10ox/DMA_L as the base case for each day, to which other cases will be compared to (Figures 2-4, (e)-(f)). The choice of this case is somewhat arbitrary, but it generally gives intermediate results relative to other simulations, as will be shown later. For each case, we set MABNAG to run until the particle reaches 40 nm in diameter or, if the mean particle growth rate is below 3.3 nm hr$^{-1}$, the model will stop after 12 hours of simulated time.

### 2.4 Growth rate calculations

#### 2.4.1 Observed growth rate (SMPS)



We have calculated the observed growth rates from the SMPS data (Figure 1, a-c). The plots for May 9 and May 11 indicate that there could be two separate nucleation events, whereas April 19 shows one event. Similarly to May 9 and May 11, the SMPS data for May 12 (Figure 1, panel c) shows what appears to be two nucleation events occurring at the surface where the SMPS collected size distributions. Tethered-balloon flight profiles for May 12

indicate that nucleation potentially occurred aloft (J. Smith, unpublished data). We hypothesize the following explanation for the "double" nucleation events observed on May 9, 11, and 12: Nucleation and growth begins to occur aloft in the residual layer. Once the mixed-layer depth grows into the residual layer, these new particles (that may have already grown to ~10 nm) then mix down and are measured at the surface. This hypothesis is supported by the presence of a high concentration of larger particles ($D_p$ = 10-30 nm) that have already undergone growth at the

"beginning" of the first event as measured by the SMPS on May 9 and May 11. Then, the second event, which presumably begins near the surface, shows a high concentration of freshly growing particles (3-5 nm, close to the limit of the SMPS detection) before larger particles appear.

As a result, we decided to calculate the growth rate based only on the second growth event for May 9 and May 11, as the second growth events are likely more representative of our ground-based measurements. For each

day, we calculated the growth rate between 10-20 nm in the following manner: we found the time at which the mode diameter hit 10, 15, and 20 nm, then used those times to calculate the mean growth rate between 10-15 nm, 15-20 nm, and 10-20 nm, using Eq. (3):

$$GR_{obs} = \frac{dDp}{dt} \cong \frac{\Delta Dp}{\Delta t} \hspace{6cm} (3)$$

This method yields a range of growth rates (Table 3) for the particles between 10-20 nm. May 9 and May 11 have higher maximum growth rates (up to 8 nm hr$^{-1}$): this could be from the influence of the continued mixing down from nucleation aloft and not actually representative of the growth rates of the particles forming near the surface.

### 2.4.2 MABNAG growth rate

MABNAG provides the wet diameter as a function of time: we calculated the rate of change of these diameters using Eq. (3) to get the modelled growth rate. Growth rates in MABNAG generally increase with size due

to the reduction of the Kelvin effect with size (gas-phase concentrations are held fixed). The growth rates generally do not change much at diameters larger than 10 nm, so we provide the average growth rate between diameters of 10-20 nm, the same range used to determine the observed growth rates.

### 2.5 HYSPLIT back trajectories

In order to assess the influence of air mass source upon each event, the NOAA HYbrid Single-Particle Lagrangian Integrated Trajectory (HYSPLIT) model (Draxler and Rolph, 2012; Rolph, 2012) with NAM meteorological data was used to obtain 48-hour air mass back trajectories (Figure 1d-f). The model was initialized at ~250 m AGL at the time of the observed NPF onset for each trajectory; a total of 24 trajectories were output for




each event day using the HYSPLIT ensemble feature that perturbs the start height by small increments vertically and horizontally.

### 3. Results

#### 3.1. April 19: Growth by organics

On April 19, 2013, a NPF event was recorded by the SMPS beginning around 12:00 Central Daylight Time (CDT) (Figure 1a); the particles grew to a diameter of about 20 nm at a rate of 3-4 nm hr$^{-1}$. The gas-phase concentrations of each measured species, averaged through this 10-20 nm diameter growth period, are presented in Table 1, and the timeseries of these observations in Figure 2a-b. Note that oxalic acid was not measured by the Cluster CIMS for this day. The ratio of measured oxalic-acid concentration to measured malonic-acid concentration was approximately 0.1 throughout the campaign when oxalic acid data was available; thus, we assume that a baseline concentration of oxalic acid was present at 0.1 times the measured concentration of malonic acid for this day. Some notable features of the gas-phase data for April 19 (Figure 2a-b) include relatively low sulfuric-acid concentrations ($\sim 2 \cdot 10^6$ cm$^{-3}$), which should only contribute to growth rates of about 0.2 nm hr$^{-1}$, or approximately 10% of the observed rates. Conversely, the concentrations of ammonia and amines are sufficiently high (100-1000 pptv) that they could play a role in sulfuric-acid neutralization and organic-salt formation. The TDCIMS particle-phase ion-fraction data (Figure 2c-d) shows primarily organics with some amines present in the particle phase, indicating that growth by acid-base reactions of organic acids and amines and/or irreversible condensation of ELVOCs is possible. As mentioned previously, we currently have no unequivocal way to distinguish between organic acids and ELVOCs or higher-volatility non-reactive organics in the TDCIMS. The organics categories presented (organics, organics with S, and organics with N) should be taken as the sum total of organics (excluding amines) detected by the TDCIMS. The TDCIMS also shows a presence of nitrate (the nitrate (ox/inorg) category) later on in the growth event. We do not expect to see significant inorganic nitric acid in the growing of sub-50 nm particles, as ammonium nitrate tends to undergo equilibrium-limited growth in submicron particles and partition proportionally to the particle mass distribution (Zhang et al., 2012b). The possibility that much of the observed ox/inorg nitrate signal arises from decomposition or ion-molecule reactions of organic nitrates cannot be excluded. Furthermore, the TDCIMS shows heightened sensitivity to inorganic nitrate with respect to sulfate (Smith et al., 2004; Lawler et al., 2014). Due to all of these uncertainties, we hesitate to attribute significant growth from inorganic nitrate.

The 48-hour HYSPLIT trajectory for April 19 (Fig 1d) shows the flow coming from the northwest. The predicted trajectories appear to be subsiding from the free troposphere over the time period and thus likely only experience surface emissions during the last 18 hours before passing through the Central Facility at Lamont, OK. The surface emissions would likely be coming from central/western Kansas, through primarily agricultural regions and no major urban areas, consistent with the low sulfuric acid concentrations. Based on these back trajectories, we hypothesize that the air mass obtained biogenic SOA precursors from the region north of the SGP site as well as high levels of gas-phase bases due to emissions from agricultural practices.





The MABNAG simulations for this day are able to corroborate the predominance of organics in the particle phase. Our base simulation, MAL/10ox/DMA_L (Fig 2e-g; Table 4) predicts a growth rate of 1.4 nm hr$^{-1}$ with 16% mole fraction from sulfuric acid; <<1% from organic acid, 24% from ammonia; 9.1% from amines; and 50% from ELVOCs. Across our sensitivity cases (Table 4), MABNAG shows negligible (<5%) amounts of organic acid in the particle phase, except for MAL_LoVP/100ox cases (an upper bound for organic-acid uptake due to lowered vapor pressure and increased gas-phase concentration), which show up to 18% of the particle was composed of organic acid. Malonic acid has a lower vapor pressure than oxalic acid, and thus more malonic acid is able to enter the particle-phase than oxalic acid. The ELVOC mole fraction tends to be around 50% for most cases, with a smaller (around 35%) mole fraction predicted for the high organic-acid cases. Since we do not know the actual contributions to growth from ELVOCs (or higher-volatility non-reactive organics) versus organic acids from the TDCIMS data, we cannot determine the accuracy of these individual species predictions. However, as the TDCIMS shows very small particle-phase contributions from bases even though high gas-phase base concentrations were also observed, this corroborates that the growth may be dominated by non-reactive organics. We see that MABNAG predicts that approximately ~16% of the particle is composed of sulfuric acid by mole (with associated ammonia). No sulfuric acid appears directly in the TDCIMS ion spectra: thus, MABNAG appears to overpredict the contribution of sulfuric acid (and associated ammonia) for this day relative to the TDCIMS ion fractions. However, since sulfuric-acid vapor concentrations were non-zero, we expect some sulfuric acid in the particle phase. The most likely reason for the discrepancy is low signal-to-noise in the TDCIMS during this period, resulting from low collected particle mass. The TDCIMS data shows some amine/amides in the particle phase: the most amine was predicted with DMA_L cases (9-11% by mole) and this compares most closely to the TDCIMS ion fractions of the amine particle-phase predictions. All TMA_L cases predict less than 1% amines by moles in the particle phase and thus likely are not realistic inputs for this day.

The MABNAG results allow us to estimate possible sulfate and organic-salt formation for this day. Figure 2g shows the mole fractions of each dissociation state of the acids and bases in addition to the mole fraction of the ELVOCs for the base simulation (MAL/10ox/DMA_L). As expected, sulfuric acid is near-fully neutralized due to the high concentrations of gas-phase bases. The amount of dissociated organic acid depends on the properties of the input acid, with oxalic acid having lower pKas than malonic acid, making oxalic acid more likely to dissociate than malonic acid (supplementary information Table S1). We see that most of the dissociated organic acids have dissociated twice, as shown in Figure 2g and supplementary information Table S1, consistent with Yli-Juuti et al, (2013). The majority of our simulations predict that less than 1% of the particle is organic acid by moles. Thus, even for the maximum organic-acid dissociation across our simulations (~90%), less than 3% of the particle (by moles) is composed of organic salts when including the bases. MAL_LoVP/100ox/DMA_L was our upper-bound simulation for organic-acid uptake; in this case a maximum ~8% (by moles) of the particle is composed of dissociated organic acid and ~23% (by moles) of the particle is composed of organic salts when including the bases. Thus, we expect the majority of growth from organics is coming from non-reactive condensing organics (ELVOCs in our simulations).

The modelled growth rate is around 1.4 nm hr$^{-1}$ for most cases with a few cases (MAL_LoVP/100ox cases) reaching up to 1.7 nm hr$^{-1}$. Thus, all cases underpredict the observed growth rates (3-4 nm hr$^{-1}$). We do note that





the organics with N and N (ox/inorg) ion categories dominate the overall TDCIMS spectrum; as MABNAG currently does not account for nitrogen-containing species beyond ammonia and amines, this could account for some of the discrepancies in the particle growth rate and composition between model and observations. As organics are a very important part of this day's particle growth, our results are sensitive to our precursor and yield assumptions of

ELVOCs, and for this day where ELVOCs dominated growth, a 50% uncertainty in ELVOC yield would correspond to close to a 50% uncertainty in growth rate (ELVOCs dominate the simulated volume fraction). Having more-direct measurements of VOCs and associated ELVOC yields will better constrain the ELVOC budget. Additionally, our lack of LVOCs, SVOCs, and accretion reactions from SVOCs also contribute to our underprediction, as these species will contribute more with increasing particle size.

10          Overall, the observations from April 19 clearly show that organic species contribute heavily to growth: the MABNAG results corroborate this, and further the MABNAG simulations show that ELVOCs dominate over organic acids for all sensitivity cases. As the TDCIMS shows small amounts of particle-phase ions from bases even though high gas-phase base concentrations were also observed, this corroborates that the growth may be dominated by non-reactive organics. Furthermore, as ELVOCs are larger molecules than the other species considered here,

their contributions to growth rates are even larger than their contribution to mole or ion fractions. Finally, we hypothesize that LVOCs, and perhaps SVOCs or accretion reactions, are contributing to growth within the 10-20 nm diameter range, as MABNAG underestimates growth without these species/reactions.

### 3.2 May 9: Growth by ammonium sulfate

On May 9, 2013 (Fig. 3), two growth events were observed; we focus our analysis on the second event, which began around 13:00 CDT. The SMPSs and Cluster CIMS both experienced instrument failure from 17:30 CDT onwards on this day; the Cluster CIMS was also not operational before 12:00 CDT. However, the two instruments captured enough of the event to inform our analysis and provide modelling inputs. By 17:30, the

particles grew to 20-40 nm diameter at a rate of 3.5-8 nm hr$^{-1}$. The Cluster CIMS measured high sulfuric acid for this day ($\sim 2 \cdot 10^7$ cm$^{-3}$), sufficiently high for sulfuric acid to contribute significantly to condensational growth. The ammonia concentrations are somewhat higher than the amine concentrations. The TDCIMS shows a high amount of ammonia and sulfate, indicating the presence of ammonium sulfate contributing strongly to the growth of the particles. A small amount of organics and amines are seen in the particle phase as well.

30          The HYSPLIT back trajectory for May 9 (Fig. 1e) shows flow from the south, through much of central/east central Texas. The predicted trajectories are entirely in the boundary layer, allowing for the possibility of the air mass experiencing surface emissions throughout the entire previous 48 hours. Many of the possible trajectories pass over or near the major metropolitan Dallas/Fort Worth region and extend into the industrial gulf-coast region; some of the trajectories extend towards the major metropolitan region of Houston. Both possible trajectory paths could

contribute SO$_2$ emissions to the air mass. Local agricultural practices could have contributed ammonia and amines to the air mass, explaining the high base concentrations present at the SGP site.

The MABNAG simulations for this day are able to capture ammonium-sulfate formation as the dominant growth pathway. Our base simulation, MAL/10ox/DMA_L, (Figure 3e-g; Table 5) predicts a growth rate of 3.2 nm





hr$^{-1}$ with 31% of the particle composition by moles from sulfuric acid; 2.2% from organic acid; 42% from ammonia; 20% from amines; and 4.3% from ELVOCs. Most sensitivity cases (Table 5) predict approximately 60-90% of the particle is composed of sulfuric acid and ammonia by mole fraction. Only the MAL_LoVP/100ox (upper bound for organic-acid uptake) cases predict otherwise; these cases show over 60% of the particle to be organic acid by moles.

However, these cases also show unrealistically high growth rates (~48-57 nm hr$^{-1}$). Based on these growth rates, we conclude that, at least for this day, growth cannot be realistically captured by the MAL_LoVP/100ox inputs; these cases will not be discussed further. The TDCIMS shows a small amount of organics and an even smaller amount of amine/amide in the particle composition. MABNAG predicts roughly 5-25% of the particle by moles to be organics (ELVOC plus organic acids) with less than 1% up to 5% of the organics by moles coming from ELVOCs. Thus,

unlike April 19, organic acid is predicted to dominate the organics contribution for this day. On a molar basis, less than 1% up to 21% of the particle is predicted to be amines.

Similar to April 19, it is seen (Figure 3g and supplementary information Table S2) that there is sufficient base to almost fully neutralize the sulfuric acid and the ratio of the remaining base to dissociated organic acid (on a molar basis) is ~0.6 to 1.9. As many cases predict negligible amounts of (<3%) organic acid by mole fraction, less

than 9% by mole of the particle is composed of organic salts. An upper limit on predicted organic salt formation comes from the case MAL_LoVP/10ox/DMA_L: following the same calculations as used for April 19, ~12.5% (by mole) of the particle is formed of organic salts when including the bases. .

MABNAG predicts growth rates between 2.9-5 nm hr$^{-1}$, with the highest growth rates seen for LoVP cases. These LoVP cases tend to predict a moderate (~15-25% by mole fraction) amount of organics (organic acid +

ELVOC) and (~<1% to 20% by mole fraction) amines in the particle phase, leading us to believe that the reduced vapor pressure of organic acids allows for the best fit simulations compared to the measurements of particle growth and composition. However, many cases still slightly underpredict the observed growth rates. This underprediction could again be from the uncertainty from the nitrogen-containing species that appear in the TDCIMS but are not accounted for in MABNAG, as well as our uncertainty in ELVOC concentrations and lack of LVOCs, SVOCs, and

accretion reactions.

Overall, the observations from May 9 show a strong contribution from ammonia and sulfate (presumably ammonium sulfate), and the MABNAG simulations corroborate this growth pathway, with the highest average mole fractions of sulfuric acid and ammonia predicted in the particle phase of the three days. This growth pathway should be well represented in regional/global models provided that emissions are well resolved.

### 3.3 May 11: Growth by sulfuric-acid/amines/organics

May 11, 2013 (Figure 4), similar to May 9, shows two growth events; we focus our analysis on the second event, which began around 15:00 CDT. All instruments were fully operational during the growth event, which is

observed to extend into May 12. The particles grow to about 25-35 nm in diameter at a rate of 3-8 nm hr$^{-1}$. The sulfuric-acid concentration on this day (~4·10$^6$ cm$^{-3}$) is in between those from the other two growth days. As with the other days, there are high ammonia and amines concentrations (100-10,000 pptv) throughout the event. The TDCIMS shows a mixed view of what is present in the particle phase during the growth event. There is a fairly



constant and significant relative amount of sulfate present in the particle. However, at the beginning of the event, amines are the dominant base present, but by 21:00, the relative amine signal has decreased and at 23:00 ammonia is dominant. Both the positive and negative signals show significant contributions from organics. The TDCIMS negative ion data also indicate the presence of nitrate; as stated previously, we hesitate to attribute significant growth from nitrate due to the unknown sensitivity of the TDCIMS to nitrate. Overall, from the TDCIMS, it appears that both sulfate and organics, as well as bases, are important for growth, but we cannot assess the relative importance of ammonia to amines for growth from the observations.

The HYSPLIT back trajectory for May 11 originates primarily from the north, travelling through central Kansas and Nebraska before reaching SGP. Some of the predicted trajectories stay in the boundary layer for the full 48 hours; others show subsidence from the free troposphere, making it difficult to assess how much of the air mass was influenced from surface emissions over the previous 48 hours. Regardless, the air mass passed through primarily agricultural regions and no major urban areas, similar to April 19, but we are unsure of the source of the sulfate on May 11.

Similar to the TDCIMS data, the MABNAG simulations for this day show varying mixtures of sulfuric acid, organics, and bases. Our base simulation, MAL/10ox/DMA_L, (Figure 4e-g; Table 6) predicts a growth rate of 0.9 nm hr$^{-1}$ with 29% of the particle composition by mole from sulfuric acid; <<1% from organic acid; 46% from ammonia; 11% from amines; and 14% from ELVOCs. Across cases, we see that roughly 10-30% by mole fraction of the particle is predicted to be sulfuric acid, in reasonable agreement with the TDCIMS data. MAL_LoVP/100ox (upper bound for organic acid uptake) cases predict up to 46% of the particle moles to be organic acid; the rest of the cases predict less than 1% up to 5% of the particle moles to be organic acid. Conversely, MABNAG predicts roughly 5-25% of the moles in the particle to be from ELVOCs, with the lowest relative ELVOC contribution seen in MAL_LoVP/100ox cases. Since the TDCIMS shows a variable amount of organics throughout the event, and we do not know the actual individual contributions from ELVOCs and organic acids, we cannot conclude which set of organics inputs best captures this day and do not exclude any set of inputs for being unrealistic. MABNAG predicts mole fractions of 35-55% for ammonia and less than 1% up to 11% for amines (with less than 1% amines predicted for all TMA cases). As the TDCIMS shows a large amount of amine/amides at the beginning of the event and a large amount of ammonia at the end of the event, we cannot determine which set of base inputs best capture this day either.

When considering organic salt formation, the majority of our simulations predict <5% of the particle by moles to be organic acid, with <15% of the particle by moles predicted to be organic salts (Figure 4g, supplementary information Table S3). The MAL_LoVP/100ox cases provide the upper limit on organic salt formation, predicting ~48% of the particle by moles to be made of organic salts. This organic-salt fraction is much greater than the upper bounds calculated for May 9 or April 11. (See Table 6, supplementary information Table S3).

Similar to April 19 and May 9, MABNAG tends to underpredict the growth rate for this day, with most cases predicting growth at around 0.9-1 nm hr$^{-1}$. The MAL_LowVP/100ox cases show slightly higher growth rates at 2.7-3.4 nm hr$^{-1}$, which does begin to capture the observed growth rates of 3-8 nm hr$^{-1}$. We reiterate our hypothesis that the underpredictions could be from the nitrogen-containing species that are detected in the TDCIMS but are not




accounted for in MABNAG, as well as our uncertainty in ELVOC concentrations and lack of LVOCs, SVOCs, and accretion reactions. Furthermore, this day shows a more variable particle-phase spectrum than April 19 or May 9, as well as a more poorly defined second growth event (Figure 1c), making the observed growth rates difficult to determine.

Overall, the observations from May 11 show that organics, sulfate, and bases (either amines or ammonia) are all important for the evolution of this new-particle growth event. The MABNAG simulations corroborate this, with the organic contribution being from ELVOCs. Growth by LVOCs and/or SVOCs, or organic accretion may also be important, as MABNAG simulations generally underestimated growth and the mole fraction of organics on this day, relative to observations. The back trajectories on this day are similar to those from April 19, though we are
unsure of the reason for the difference in sulfuric acid concentrations between the two days. Similar to April 19, the TDCIMS tends to show more organics than bases that would remain after neutralizing the observed particle-phase sulfuric acid, corroborating that the organics in the particle phase are likely dominated by non-reactive organics.

**3.4 Synthesis across days**

For the 3 days analyzed here, new-particle growth at SGP can be driven by combinations of sulfuric acid (with associated bases) and non-reactive organics, of which ELVOCs contribute a substantial fraction (at least for the yields assumed here). The exact mixture of these pathways depends on the airmass history. We found that the contribution of small organics and organic salts, such as oxalic and malonic acid and associated salts formed with
ammonia and amines, to growth may be minor at SGP. However, decreasing the assumed vapor pressure and/or increasing the vapor-phase concentration of the organic acids (within uncertainty ranges) increased the contribution of the small organic acids on some days. Both modelling and measurements show that both ammonia and amines can act as the bases in growing nanoparticles at SGP. While the MABNAG simulations here are limited in the number of species and growth-processes considered, the model is capable of qualitatively differentiating the
dominant particle-phase compositions between the 3 days: organics on April 19, inorganics on May 9, and a mixture on May 11.

Although not discussed above, we also considered the effects of RH uncertainty on our results: April 19 and May 11 both have much lower relative humidities (32% and 36%, respectively) than May 9 (69%). MABNAG shows a moderate sensitivity to RH. We ran a simulation of all days and all cases at 80% RH (not shown); the
simulations showed an increase in the dissociation of both malonic and oxalic acids as well as an increase in growth rate for all cases (in part due to increased water uptake), with most cases showing an associated increase in the mole fraction of organic acid. The increase in growth rate depended on the organic acid concentration input and vapor pressure, with the highest increases seen for LoVP/100ox cases. These higher-RH results may be applicable since the boundary layer was well-mixed on the three nucleation days, and the RH increases with height within well-
mixed boundary layers. Thus, using surface-based measurements for RH may be a lower-bound for RH and cause growth underestimates.

**3.5 Limitations of This Study**



While we have shown that MABNAG can quantitatively capture the dominant species that contribute to growth for observed growth events, this study is limited in its scope due to the following uncertainties and limitations:

- There are significant uncertainties in both the measured organic-acid concentrations and chemical properties. The measured oxalic-acid concentrations could be up to 100 times too low due to the uncertainty in the oxalic-acid detection efficiency in the Cluster CIMS. Also, the malonic acid sensitivity is not known. The saturation vapor pressures of malonic and oxalic acid show variation amongst the reported values, and our simulation results are sensitive to their vapor pressures within the reported ranges.

- There is not yet a constrained ELVOC budget from the oxidation of atmospheric VOCs. The yields from different species and under different atmospheric conditions are just beginning to be quantified. The fixed, 3% yield that we used here is preliminary and must be refined as the community continues to learn more about ELVOCs. The confidence in our estimated ELVOC budget also is limited by uncertainties stemming from using MEGAN output for the monoterpene-concentration estimate, and by uncertainties in the local ozone concentrations.

- Large (greater than C3) amines are relatively unstudied in the field as of yet, and the exact identification of these molecules is difficult with current instrumentation. As a result, estimating the thermodynamic properties such as pKa and vapor pressure that determine abilities of these amines to participate in acid-base reactions is difficult, and we can only provide estimates of these contributions.

- Our particle-phase composition measurements from the TDCIMS provide only qualitative information for the organic species present in the growing particles. We do not know the exact molar contributions to the particles from each species, as the TDCIMS is not calibrated for each of the many organic compounds that are detected due to fragmentation during desorption as well as chemical ionization of desorbed gas phase ions. Perhaps more significantly, particle phase "matrix effects" may impact the efficiency by which organic compounds are desorbed and ionized; such matrix effects are difficult to assess since they depend on the coexisting compounds in the particles and the phase of the particles.

- We did not know the parent molecule(s) of the nitrate signal in the TDCIMS ions that is classified as either inorganic or oxidized nitrate. This signal appears non-trivially during part of every growth event analyzed, but we are without knowledge of its origin.

- The MABNAG model, as used here, only simulates one organic acid and one amine in any individual simulation. This limits our ability to determine the contribution of combinations of organic acids and amines to growth through acid-base reactions and condensation (for the less-volatile organic acids). Instead, we present only limiting cases that inform us of the potential contributions of organic acids and amines if the sum of oxalic and malonic acid had the properties of one these species.

- We did not account for the contribution of LVOCs or SVOCs to condensational growth as there were no gas concentration measurements of such compounds. As particles grow beyond initial cluster sizes, the LVOCs will begin to contribute to growth, and likely are a significant contributor for particles as they approach diameters of 10 nm (Pierce et al., 2011). As the particle continues to grow, the SVOCs may also



be a non-trivial contributor to growth (Pierce et al., 2011). Thus, the growth by non-reactive organics is likely underestimated in this study.

- We did not account for accretion reactions that could contribute to particle growth as there were no observations to constrain the contribution of accretion products to new-particle growth during this study. Accretion has been observed in the laboratory in particles greater than 4 nm in size (Wang et al., 2010) and thus has the potential to contribute to growth even at these smaller particle sizes.

- We assumed in MABNAG that all species in the particle phase instantaneously homogeneously mix into a liquid phase: this assumption ignores any particle-phase diffusion limitations that can arise from heterogeneous particle phases. It's estimated that such diffusion limitations can begin to matter at particle sizes greater than 20 nm in diameter.

- We use RH measured at the surface, which may be an underestimate of RH in other portions of a well-mixed boundary layer. MABNAG sensitivity simulations with increased RH showed increased growth rates and contributions from organic acids.

## 4. Conclusions

In this study, we sought to understand the species/mechanisms that contribute to the growth of newly formed particles at the US Department of Energy Atmospheric Radiation Measurement program Southern Great Plains (SGP) field site in Oklahoma, US, and to find closure in particle growth rates and composition between the SGP measurements and the growth model, MABNAG. We analyzed data collected from April 13-May 25, 2013 for the SGP New Particle Formation Study (NPFS). We focused the analysis on three new-particle formation and growth events occurring on April 19, May 9, and May 11. These days had different dominant species contributing to growth: April 19 was primarily from organics, May 9 was from ammonium sulfate, and May 11 was from organics, amines/ammonia, and sulfate. MABNAG was constrained by the measured gas-phase concentrations of key atmospheric species present during the growth event for each day, and we found that MABNAG qualitatively simulated the observed dominant species for each day under certain sets of assumptions. We saw that during the NPFS campaign, new-particle growth events can be explained by either sulfuric acid forming salts with atmospheric bases (either ammonia or amines) or the condensation of primarily non-reactive organics, or a combination of these two. MABNAG can qualitatively capture different dominant growth pathways. It appears from the TDCIMS that most of the organics measured are likely non-reactive: if we assume equivalent detection efficiencies, there are generally more organics than there are bases. The MABNAG simulations support that the organics in the growing particles are likely non-reactive, with the non-reactive-organic ELVOC input species dominating the organic contribution to the particle growth over the organic acid input species in almost every sensitivity case.

MABNAG tends to underpredict the observed growth rates. Due to the strong organics signals in the TDCIMS, we propose that these low growth rates are mainly due to an underrepresentation of organic uptake in MABNAG, either by non-reactive condensation of LVOCs or SVOCs or particle-phase accretion. Furthermore, the discovery of ELVOCs is relatively new and the ELVOC budget remains largely unconstrained.




Although we have not achieved complete closure in particle growth rates and composition between the SGP measurements and MABNAG simulations, we present this work as an important step towards understanding new-particle formation and growth events. We find that the relatively poorly understood ELVOC species can play a key role in the growth of particles through non-reactive condensation. Yet, organics of higher but still sufficiently low vapor pressures ($\sim$<$10^0$ µg m$^{-3}$ saturation mass concentration) are likely also important for growth, and increase in importance with increasing particle size. Based on these findings, we encourage more field-based measurements that focus on the speciation and properties of organics, both in the gas phase and in particles. In particular, gas-phase ELVOC, LVOC, and SVOC measurements, found either through speciation or volatility measurements, would greatly inform future modelling efforts. These measurements are exceedingly challenging but as experimental techniques evolve, they will be invaluable in understanding and modelling both aerosol fundamentals and aerosol impacts on climate and human health.

**Acknowledgements**

This research was supported by the U.S. Department of Energy's Atmospheric System Research, an Office of Science, Office of Biological and Environmental Research program, under Grant No. DE-SC0011780 and Grant No. DE-SC0014469. U.S. Department of Energy as part of the Atmospheric Radiation Measurement (ARM) Climate Research Facility [XXX sites, data, specific campaign data] were used. Coty Jen was supported under a NSF AGS Postdoctoral Fellowship. Taina Yli-Juuti was supported by the Academy of Finland Centre of Excellence (grant no. 272041) and strategic funding from University of Eastern Finland.



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

Appendix A: List of abbreviations

| | |
|---|---|
| AGL | Above Ground Level |
| AmPMS | Ambient pressure Proton transfer Mass Spectrometer |
| ARM | Atmospheric Radiation Measurement |
| BL | Boundary Layer |
| BVOC | Biological Volatile Organic Compound |
| CCN | Cloud Condensation Nuclei |
| CDT | Central Daylight Time |
| CIMS | Chemical Ionization Mass Spectrometer |
| CPC | Condensation Particle Counter |
| CS | Condensation Sink |
| DMA | Dimethylamine |
| DOE | Department of Energy |
| $D_p$ | Particle Diameter |
| E-AIM | Extended Aerosol Inorganics Model |
| EL-VOC | Extremely Low-Volatility Organic Compound |
| EPA | Environmental Protection Agency |
| GR | Growth Rate |
| HYSPLIT | HYbrid Single-Particle Lagrangian Integrated Trajectory |
| IUPAC | International Union of Pure and Applied Chemistry |
| L | sum of Light (C1-C3) amines measured at SGP |
| Lo-VP | reducing the vapor pressure of the organic acid input in MABNAG by $10^{-1}$ |
| L-VOC | Low-Volatile Organic Compound |



|   |        |                                                            |
|---|--------|------------------------------------------------------------|
|   | MABNAG | Model for Acid-Base chemistry in NAnoparticle Growth       |
|   | MAL    | Malonic acid                                               |
|   | MEGAN  | Model of Emissions of Gases and Aerosols in Nature         |
|   | NAM    | North American Mesoscale model                             |
| 5 | NOAA   | National Oceanic and Atmospheric Administration            |
|   | NPF    | New-Particle Formation                                     |
|   | NPFS   | New-Particle Formation Study                               |
|   | OX     | Oxalic acid                                                |
|   | PTR-MS | Proton Transfer Reaction-Mass Spectrometer                 |
| 10 | RH    | Relative Humidity                                          |
|   | SGP    | Southern Great Plains                                      |
|   | SMPS   | Scanning Mobility Particle Sizer                           |
|   | SOA    | Secondary Organic Aerosol                                  |
|   | S-VOC  | Semi-Volatile Organic Compound                             |
| 15 | T     | sum of Total amines measured at SGP                        |
|   | TDCIMS | Thermal Decomposition Chemical Ionization Mass Spectrometer |
|   | TMA    | Trimethylamine                                             |
|   | UNIFAC | UNIquac Functional-group Activity Coefficient method       |
|   | VOC    | Volatile Organic Compounds                                 |
| 20 |       |                                                            |



Table 1. Gas-phase concentration and temperature inputs to MABNAG for each day.

| Day | T [C] | RH (%) | Sulfuric Acid [cm$^{-3}$] | Organic Acid: mal+ox / mal+10ox / mal + 100ox [cm$^{-3}$] | Ammonia [cm$^{-3}$] | Amine: Light / Total [cm$^{-3}$] | EL-VOC [cm$^{-3}$] |
|---|---|---|---|---|---|---|---|
| 4/19/13 | 11.6 | 32 | $2.4 \cdot 10^6$ | $1.17 \cdot 10^7$ / $2.17 \cdot 10^7$ / $1.10 \cdot 10^8$ | $2.98 \cdot 10^{10}$ | $2.91 \cdot 10^9$ / $4.8 \cdot 10^{10}$ | $1.22 \cdot 10^7$ |
| 5/09/13 | 12.7 | 69 | $1.97 \cdot 10^7$ | $7.15 \cdot 10^7$ / $1.49 \cdot 10^8$ / $9.11 \cdot 10^8$ | $8.94 \cdot 10^9$ | $1.01 \cdot 10^9$ / $2.41 \cdot 10^{10}$ | $4.3 \cdot 10^6$ |
| 5/11/13 | 16.4 | 36 | $5.3 \cdot 10^6$ | $2.66 \cdot 10^7$ / $6.43 \cdot 10^7$ / $4.14 \cdot 10^8$ | $1.11 \cdot 10^{10}$ | $1.54 \cdot 10^9$ / $1.85 \cdot 10^{10}$ | $4.1 \cdot 10^6$ |



Table 2. Chemical properties for each species input in MABNAG.

| Species | Molar Mass [g/mol] | pKa 1 | pKa 2 | Vapor Pressure of pure compound [Pa] (µg m-3) | Henry's Law Constant [mol kg-1 atm-1] | Diffusion Coefficient [m2 s-1] |
|---|---|---|---|---|---|---|
| Sulfuric Acid | 98.1 | -3 | 1.99 | 0 | n/a | $9.4 \cdot 10^{-6}$ |
| Malonic Acid | 104.1 | 2.85 | 5.7 | $4 \cdot 10^{-5}$ ($1.8 \cdot 10^{-5}$) | n/a | $8.4 \cdot 10^{-6}$ [a] |
| Oxalic Acid | 90.03 | 1.46 | 4.4 | $4 \cdot 10^{-3}$ ($1.5 \cdot 10^{-3}$) | n/a | $8.4 \cdot 10^{-6}$ [b] |
| Ammonia | 17.03 | 9.25 | n/a | n/a | 60.7[c] | n\a |
| DMA | 45.1 | 10.7 | n/a | n/a | 31.41[d] | n\a |
| TMA | 59.1 | 9.8 | n/a | n/a | 9.6[d] | n\a |
| ELVOC[e] | 280 | n/a | n/a | $1 \cdot 10^{-9}$ ($1.2 \cdot 10^{-3}$) | n/a | $5 \cdot 10^{-6}$ |

[a] Calculated using the Fuller et al., method (Eq. 11-4.4 in Poling et al., 2014)

[b] Assumed to be the same as malonic acid

5    [c] Haar and Gallagher, 1978

[d] http://webbook.nist.gov/chemistry/

[e] Assumed properties of the ELVOC species



Table 3. Observed growth-rate ranges for each day.

| Day | Growth Rate [nm hr$^{-1}$] |
| --- | --- |
| April 19 | 3-4 |
| May 9 | 3.5-8 |
| May 11 | 3-8 |





Table 4. Modelled Growth Rates and final mole fractions for April 19. The base case (MAL/10ox/DMA_L) is in bold. The observed growth rate for this day is 2-4 nm hr$^{-1}$.

| Case | Growth Rate [nm hr$^{-1}$] | Sulfuric Acid mole fraction | Organic Acid mole fraction | Ammonia mole fraction | Amine mole fraction | ELVOC mole fraction |
|------|------|------|------|------|------|------|
| MAL/1ox/DMA_L | 1.4 | 0.16 | 0.0027 | 0.24 | 0.09 | 0.5 |
| **MAL/10ox/DMA_L** | **1.4** | **0.16** | **0.0039** | **0.24** | **0.091** | **0.5** |
| MAL/100ox/DMA_L | 1.4 | 0.16 | 0.024 | 0.24 | 0.092 | 0.48 |
| MAL/1ox/TMA_L | 1.4 | 0.16 | 0.0021 | 0.32 | 0.0042 | 0.51 |
| MAL/10ox/TMA_L | 1.4 | 0.16 | 0.039 | 0.32 | 0.0042 | 0.5 |
| MAL/100ox/TMA_L | 1.4 | 0.16 | 0.019 | 0.32 | 0.0042 | 0.49 |
| MAL/1ox/TMA_T | 1.4 | 0.16 | 0.0024 | 0.27 | 0.058 | 0.51 |
| MAL/10ox/TMA_T | 1.4 | 0.16 | 0.0044 | 0.27 | 0.058 | 0.5 |
| MAL/100ox/TMA_T | 1.4 | 0.16 | 0.022 | 0.27 | 0.059 | 0.49 |
| OX/1ox/DMA_L | 1.4 | 0.16 | 0.0028 | 0.24 | 0.11 | 0.51 |
| OX/10ox/DMA_L | 1.4 | 0.16 | 0.0008 | 0.24 | 0.09 | 0.51 |
| OX/100ox/DMA_L | 1.4 | 0.16 | 0.004 | 0.24 | 0.091 | 0.5 |
| OX/1ox/TMA_L | 1.4 | 0.16 | 0.000044 | 0.32 | 0.004 | 0.5 |
| OX/10ox/TMA_L | 1.4 | 0.16 | 0.000058 | 0.32 | 0.004 | 0.5 |
| OX/100ox/TMA_L | 1.4 | 0.16 | 0.0023 | 0.32 | 0.0042 | 0.5 |
| OX/1ox/TMA_T | 1.4 | 0.16 | 0.000049 | 0.27 | 0.058 | 0.51 |
| OX/10ox/TMA_T | 1.4 | 0.16 | 0.00063 | 0.27 | 0.058 | 0.51 |
| OX/100ox/TMA_T | 1.4 | 0.16 | 0.0032 | 0.27 | 0.058 | 0.5 |
| MAL_LoVP/1ox/DMA_L | 1.4 | 0.16 | 0.02 | 0.24 | 0.09 | 0.48 |
| MAL_LoVP/10ox/DMA_L | 1.5 | 0.15 | 0.04 | 0.25 | 0.09 | 0.46 |
| MAL_LoVP/100ox/DMA_L | 1.7 | 0.11 | 0.18 | 0.27 | 0.1 | 0.34 |
| MAL_LoVP/1ox/TMA_L | 1.4 | 0.16 | 0.025 | 0.33 | 0.092 | 0.49 |
| MAL_LoVP/10ox/TMA_L | 1.4 | 0.16 | 0.045 | 0.33 | 0.094 | 0.48 |
| MAL_LoVP/100ox/TMA_L | 1.6 | 0.12 | 0.18 | 0.34 | 0.1 | 0.37 |
| MAL_LoVP/1ox/TMA_T | 1.4 | 0.16 | 0.02 | 0.27 | 0.0042 | 0.49 |
| MAL_LoVP/10ox/TMA_T | 1.4 | 0.15 | 0.036 | 0.28 | 0.0042 | 0.47 |





| | | | | | | |
|---|---|---|---|---|---|---|
| MAL_LoVP/100ox/TMA_T | 1.7 | 0.11 | 0.16 | 0.29 | 0.0044 | 0.35 |
| OX_LoVP/1ox/DMA_L | 1.4 | 0.16 | 0.023 | 0.24 | 0.059 | 0.5 |
| OX_LoVP/10ox/DMA_L | 1.4 | 0.16 | 0.041 | 0.24 | 0.059 | 0.5 |
| OX_LoVP/100ox/DMA_L | 1.4 | 0.15 | 0.17 | 0.26 | 0.063 | 0.46 |
| OX_LoVP/1ox/TMA_L | 1.4 | 0.16 | 0.0042 | 0.32 | 0.091 | 0.5 |
| OX_LoVP/10ox/TMA_L | 1.4 | 0.16 | 0.0077 | 0.33 | 0.092 | 0.5 |
| OX_LoVP/100ox/TMA_L | 1.4 | 0.16 | 0.034 | 0.34 | 0.099 | 0.48 |
| OX_LoVP/1ox/TMA_T | 1.4 | 0.16 | 0.0024 | 0.27 | 0.0042 | 0.5 |
| OX_LoVP/10ox/TMA_T | 1.4 | 0.16 | 0.0044 | 0.27 | 0.0042 | 0.5 |
| OX_LoVP/100ox/TMA_T | 1.4 | 0.15 | 0.021 | 0.29 | 0.0044 | 0.47 |



Table 5. Modelled Growth Rates and final mole fractions for May 9. The base case (MAL/10ox/DMA_L) is in bold. The observed growth rate for this day is 4-10 nm hr$^{-1}$.

| Case | Growth Rate [nm hr$^{-1}$] | Sulfuric Acid mole fraction | Organic Acid mole fraction | Ammonia mole fraction | Amine mole fraction | ELVOC mole fraction |
|---|---|---|---|---|---|---|
| MAL/1ox/DMA_L | 3.1 | 0.32 | 0.01 | 0.42 | 0.2 | 0.044 |
| **MAL/10ox/DMA_L** | **3.2** | **0.31** | **0.022** | **0.42** | **0.2** | **0.043** |
| MAL/100ox/DMA_L | 4.2 | 0.26 | 0.15 | 0.37 | 0.18 | 0.036 |
| MAL/1ox/TMA_L | 2.9 | 0.32 | 0.0096 | 0.61 | 0.0099 | 0.045 |
| MAL/10ox/TMA_L | 3 | 0.32 | 0.02 | 0.61 | 0.0098 | 0.044 |
| MAL/100ox/TMA_L | 3.8 | 0.27 | 0.14 | 0.54 | 0.0088 | 0.038 |
| MAL/1ox/TMA_T | 3.2 | 0.32 | 0.01 | 0.45 | 0.18 | 0.044 |
| MAL/10ox/TMA_T | 3.2 | 0.31 | 0.022 | 0.44 | 0.18 | 0.044 |
| MAL/100ox/TMA_T | 4.2 | 0.26 | 0.15 | 0.4 | 0.16 | 0.036 |
| OX/1ox/DMA_L | 3.1 | 0.32 | 0.00075 | 0.43 | 0.21 | 0.045 |
| OX/10ox/DMA_L | 3.1 | 0.32 | 0.0016 | 0.43 | 0.21 | 0.045 |
| OX/100ox/DMA_L | 3.1 | 0.31 | 0.0096 | 0.43 | 0.21 | 0.044 |
| OX/1ox/TMA_L | 2.9 | 0.33 | 0.00043 | 0.62 | 0.01 | 0.045 |
| OX/10ox/TMA_L | 2.9 | 0.33 | 0.0009 | 0.62 | 0.01 | 0.045 |
| OX/100ox/TMA_L | 2.9 | 0.32 | 0.0055 | 0.62 | 0.01 | 0.045 |
| OX/1ox/TMA_T | 3.1 | 0.32 | 0.00068 | 0.45 | 0.18 | 0.045 |
| OX/10ox/TMA_T | 3.1 | 0.32 | 0.0014 | 0.45 | 0.18 | 0.045 |
| OX/100ox/TMA_T | 3.2 | 0.32 | 0.0087 | 0.45 | 0.18 | 0.044 |
| MAL_LoVP/1ox/DMA_L | 3.8 | 0.28 | 0.1 | 0.39 | 0.19 | 0.039 |
| MAL_LoVP/10ox/DMA_L | 5 | 0.22 | 0.22 | 0.35 | 0.17 | 0.032 |
| MAL_LoVP/100ox/DMA_L | 56.8 | 0.024 | 0.63 | 0.23 | 0.11 | 0.0048 |
| MAL_LoVP/1ox/TMA_L | 3.4 | 0.29 | 0.095 | 0.57 | 0.0092 | 0.04 |
| MAL_LoVP/10ox/TMA_L | 4.4 | 0.24 | 0.21 | 0.5 | 0.0082 | 0.034 |
| MAL_LoVP/100ox/TMA_L | 48.6 | 0.028 | 0.69 | 0.27 | 0.0045 | 0.0054 |
| MAL_LoVP/1ox/TMA_T | 3.8 | 0.28 | 0.1 | 0.42 | 0.16 | 0.039 |
| MAL_LoVP/10ox/TMA_T | 5 | 0.23 | 0.22 | 0.37 | 0.15 | 0.032 |





| | | | | | |
|---|---|---|---|---|---|
| MAL_LoVP/100ox/TMA_T | 56 | 0.025 | 0.64 | 0.24 | 0.096 | 0.0049 |
| OX_LoVP/1ox/DMA_L | 3.1 | 0.32 | 0.0075 | 0.43 | 0.21 | 0.044 |
| OX_LoVP/10ox/DMA_L | 3.2 | 0.31 | 0.016 | 0.42 | 0.21 | 0.043 |
| OX_LoVP/100ox/DMA_L | 4.2 | 0.24 | 0.097 | 0.42 | 0.21 | 0.034 |
| OX_LoVP/1ox/TMA_L | 2.9 | 0.32 | 0.0043 | 0.62 | 0.01 | 0.045 |
| OX_LoVP/10ox/TMA_L | 2.9 | 0.32 | 0.009 | 0.62 | 0.01 | 0.044 |
| OX_LoVP/100ox/TMA_L | 3.4 | 0.28 | 0.056 | 0.61 | 0.01 | 0.04 |
| OX_LoVP/1ox/TMA_T | 3.2 | 0.32 | 0.0068 | 0.45 | 0.18 | 0.044 |
| OX_LoVP/10ox/TMA_T | 3.2 | 0.31 | 0.014 | 0.45 | 0.18 | 0.043 |
| OX_LoVP/100ox/TMA_T | 4.1 | 0.25 | 0.088 | 0.45 | 0.18 | 0.035 |





Table 6. Modelled Growth Rates and final mole fractions for May 11. The base case (MAL/10ox/DMA_L) is in bold. The observed growth rate for this day is 5-10 nm hr$^{-1}$.

| Case | Growth Rate [nm hr$^{-1}$] | Sulfuric Acid mole fraction | Organic Acid mole fraction | Ammonia mole fraction | Amine mole fraction | ELVOC mole fraction |
|---|---|---|---|---|---|---|
| MAL/1ox/DMA_L | 0.9 | 0.29 | 0.0023 | 0.46 | 0.11 | 0.14 |
| **MAL/10ox/DMA_L** | **0.9** | **0.29** | **0.0056** | **0.46** | **0.11** | **0.14** |
| MAL/100ox/DMA_L | 1 | 0.27 | 0.039 | 0.44 | 0.11 | 0.13 |
| MAL/1ox/TMA_L | 0.9 | 0.29 | 0.0022 | 0.56 | 0.0048 | 0.14 |
| MAL/10ox/TMA_L | 0.9 | 0.29 | 0.0052 | 0.56 | 0.0047 | 0.14 |
| MAL/100ox/TMA_L | 0.9 | 0.28 | 0.037 | 0.55 | 0.0046 | 0.13 |
| MAL/1ox/TMA_T | 0.9 | 0.29 | 0.0022 | 0.52 | 0.053 | 0.14 |
| MAL/10ox/TMA_T | 0.9 | 0.29 | 0.0054 | 0.52 | 0.053 | 0.14 |
| MAL/100ox/TMA_T | 1 | 0.28 | 0.038 | 0.5 | 0.051 | 0.13 |
| OX/1ox/DMA_L | 0.9 | 0.29 | 0.0039 | 0.46 | 0.082 | 0.24 |
| OX/10ox/DMA_L | 0.9 | 0.29 | 0.00042 | 0.46 | 0.11 | 0.14 |
| OX/100ox/DMA_L | 0.9 | 0.29 | 0.0029 | 0.46 | 0.11 | 0.14 |
| OX/1ox/TMA_L | 0.9 | 0.29 | 0.000043 | 0.56 | 0.0032 | 0.2 |
| OX/10ox/TMA_L | 0.9 | 0.29 | 0.0003 | 0.56 | 0.0048 | 0.14 |
| OX/100ox/TMA_L | 0.9 | 0.29 | 0.0021 | 0.56 | 0.0048 | 0.14 |
| OX/1ox/TMA_T | 0.9 | 0.29 | 0.000046 | 0.52 | 0.036 | 0.2 |
| OX/10ox/TMA_T | 0.9 | 0.29 | 0.00035 | 0.52 | 0.053 | 0.14 |
| OX/100ox/TMA_T | 0.9 | 0.29 | 0.0024 | 0.52 | 0.053 | 0.14 |
| MAL_LoVP/1ox/DMA_L | 1 | 0.28 | 0.023 | 0.45 | 0.11 | 0.13 |
| MAL_LoVP/10ox/DMA_L | 1 | 0.27 | 0.057 | 0.44 | 0.11 | 0.13 |
| MAL_LoVP/100ox/DMA_L | 3.4 | 0.1 | 0.46 | 0.31 | 0.079 | 0.048 |
| MAL_LoVP/1ox/TMA_L | 0.9 | 0.28 | 0.022 | 0.56 | 0.0047 | 0.14 |
| MAL_LoVP/10ox/TMA_L | 1 | 0.27 | 0.053 | 0.54 | 0.0046 | 0.13 |
| MAL_LoVP/100ox/TMA_L | 2.7 | 0.12 | 0.45 | 0.37 | 0.0032 | 0.057 |
| MAL_LoVP/1ox/TMA_T | 0.9 | 0.28 | 0.022 | 0.51 | 0.052 | 0.14 |
| MAL_LoVP/10ox/TMA_T | 1 | 0.27 | 0.055 | 0.5 | 0.051 | 0.13 |





| | | | | | |
|---|---|---|---|---|---|
| MAL_LoVP/100ox/TMA_T | 2.9 | 0.11 | 0.46 | 0.34 | 0.036 | 0.054 |
| OX_LoVP/1ox/DMA_L | 0.9 | 0.29 | 0.0017 | 0.46 | 0.11 | 0.14 |
| OX_LoVP/10ox/DMA_L | 0.9 | 0.29 | 0.0042 | 0.46 | 0.11 | 0.14 |
| OX_LoVP/100ox/DMA_L | 1 | 0.27 | 0.028 | 0.46 | 0.11 | 0.13 |
| OX_LoVP/1ox/TMA_L | 0.9 | 0.29 | 0.0012 | 0.56 | 0.0048 | 0.14 |
| OX_LoVP/10ox/TMA_L | 0.9 | 0.29 | 0.003 | 0.56 | 0.0048 | 0.14 |
| OX_LoVP/100ox/TMA_L | 0.9 | 0.28 | 0.02 | 0.57 | 0.0048 | 0.13 |
| OX_LoVP/1ox/TMA_T | 0.9 | 0.29 | 0.0014 | 0.52 | 0.053 | 0.14 |
| OX_LoVP/10ox/TMA_T | 0.9 | 0.29 | 0.0035 | 0.52 | 0.053 | 0.14 |
| OX_LoVP/100ox/TMA_T | 1 | 0.27 | 0.023 | 0.52 | 0.053 | 0.13 |





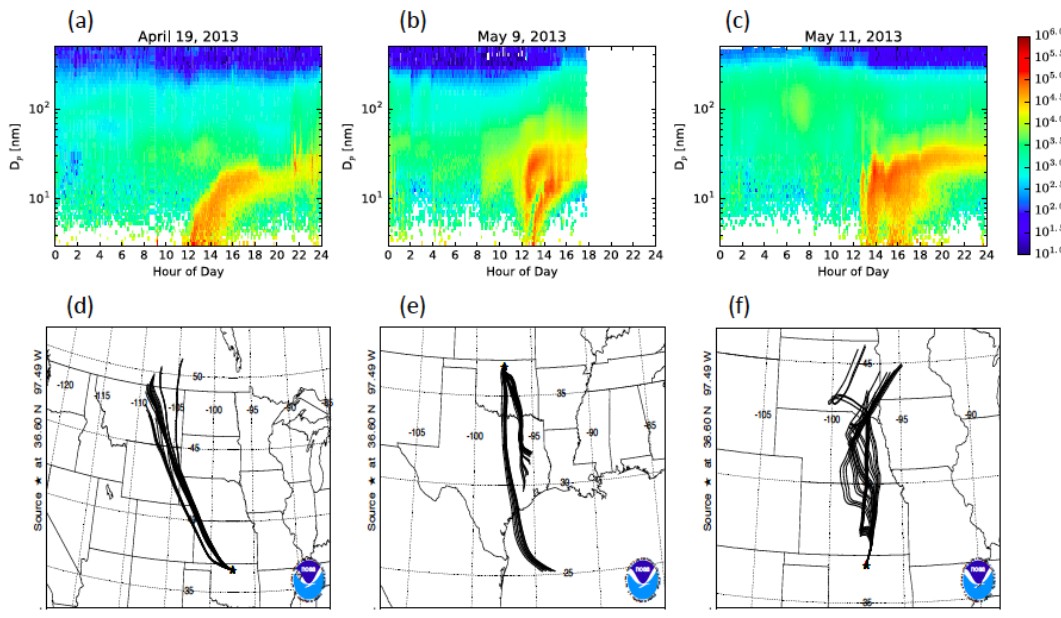

Figure 1: (a-c) The growth events for April 19, May 09, and May 11, 2013, as captured by a scanning mobility particle sizer at SGP. Each plot shares the same colorbar. (d-f) The associated 48 hour Hysplit back trajectories for each day as calculated using the NOAA Hysplit Model with NAM meteorological data, initialized at 250 m AGL.





Figure 2: Measurements and MABNAG predictions for the base-case simulation, MAL/10ox/DMA_L, for April 19, 2013. (a) Gas-phase acids and ELVOC estimate. Oxalic acid was not measured for this day; the cluster CIMS was not operational before 9:00 CDT for this day. (b) Gas phase bases. (c)-(d) Particle-phase data. The TDCIMS was not operational before 14:00 CDT (e) Size distribution from the three merged SMPSs with the modeled growth rate (black line). Overlaid is the mean collection diameter from the TDCIMS for the positive (red points) and negative (black points) signals . (f)-(g) Modelled particle composition as a function of size. (f) shows the lumped mole





fractions (excluding water) of each species, including any dissociation products. (g) shows the individual mole fractions of each species and its dissociation products. $NH_3$ and DMA are not shown as both species dissociate almost entirely to $NH_4^+$ and $DMA^+$, respectively. OH- is not shown as its concentration is extremely low (~$10^{-15}$).

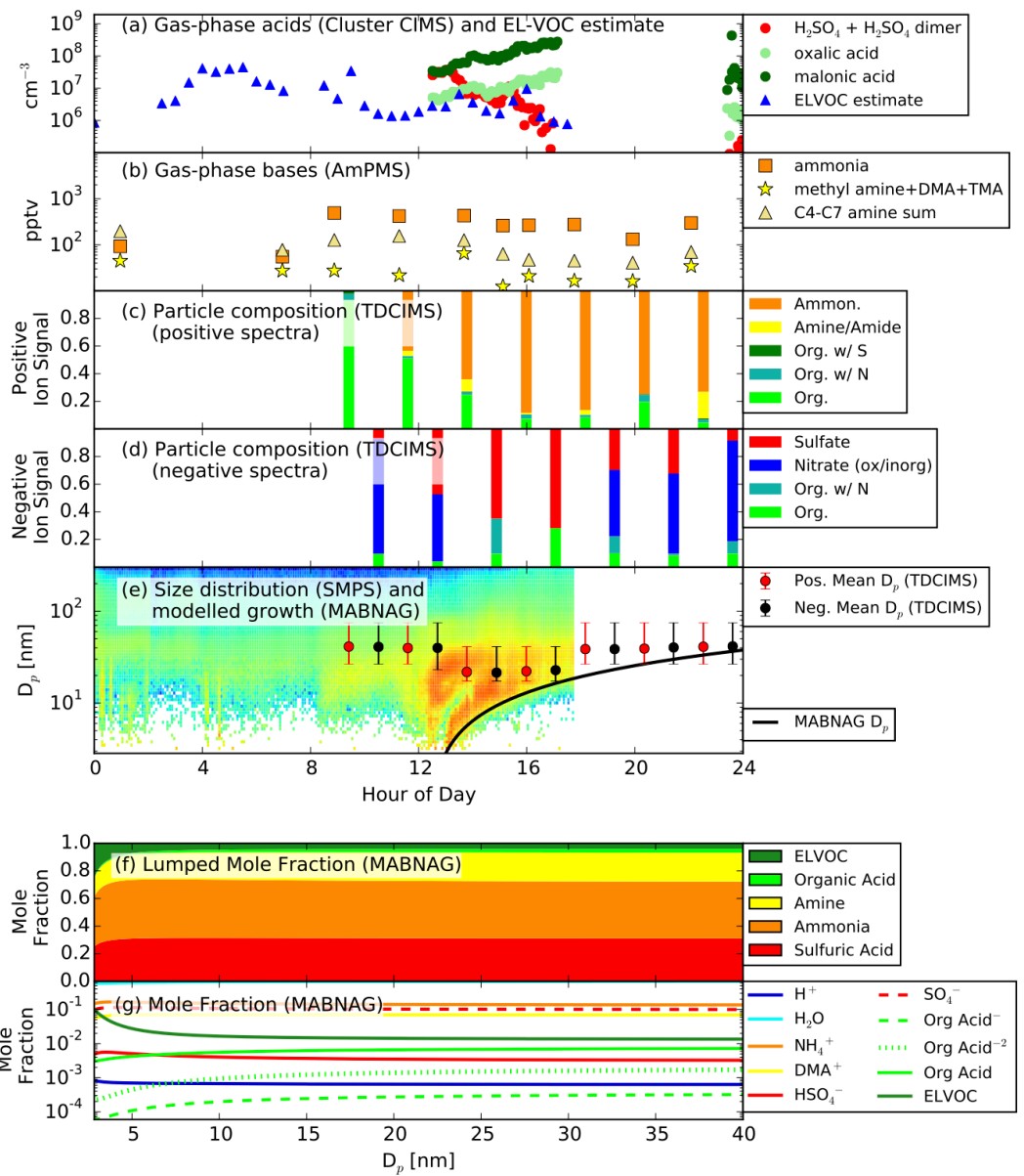

Figure 3. Measurements and MABNAG predictions for the base-case simulation, MAL/10ox/DMA_L, for May 9,
2013. (a) Gas-phase acids and ELVOC estimate. The Cluster CIMS was not operational between 17:30-23:00 CDT
for this day. (b) Gas phase bases. The AmPMS was not operational between 2:00-6:00 CDT for this day.(c)-(d)
Particle-phase data. The TDCIMS was not operational before 9:00 CDT for this day. (e) Size distribution from the





three merged SMPSs with the modeled growth rate (black line). Overlaid is the mean collection diameter from the TDCIMS for the positive (red points) and negative (black points) signals. The SMPSs were not operational after 17:30 CDT for this day. (f)-(g) Modelled particle composition as a function of size. (f) shows the lumped mole fractions (excluding water) of each species, including any dissociation products. (g) shows the individual mole

5      fractions of each species and its dissociation products. $NH_3$ and DMA are not shown as both species dissociate almost entirely to $NH_4^+$ and $DMA^+$, respectively. OH- is not shown as its concentration is extremely low ($\sim 10^{-15}$).





Figure 4: Measurements and MABNAG predictions for the base-case simulation, MAL/10ox/DMA_L, for May 11, 2013. (a) Gas-phase acids and ELVOC estimate. (b) Gas phase bases. (c)-(d) Particle-phase data. (e) Size distribution from the three merged SMPSs with the modeled growth rate (black line). Overlaid is the mean collection diameter from the TDCIMS for the positive (red points) and negative (black points) signals . (f)-(g) Modelled particle composition as a function of size. (f) shows the lumped mole fractions (excluding water) of each species, including any dissociation products. (g) shows the individual mole fractions of each species and its





dissociation products. NH$_3$ and DMA are not shown as both species dissociate almost entirely to NH$_4^+$ and DMA$^+$, respectively. OH- is not shown as its concentration is extremely low (~10$^{-15}$).

