# Peer review of "Multiple new-particle growth pathways observed at the US DOE Southern Great Plains field site"

_Atmospheric Chemistry and Physics, 2016_

## Referee Comment (RC1) · Anonymous Referee #3 · 15 Apr 2016

Review of **Analysis of multiple new-particle growth pathways observed at the US DOE Southern Great Plains field site by** Hodshire et al.

This article adds to the understanding of particle growth processes in the atmosphere. The article is well written and the topic fits to the scope of the journal. The article gives valuable, yet qualitative information about the growth pathways of nano-particles, which is derived from both measurement and modelling. I find it especially important that the authors show that the major growth mechanism can vary even at the same site and the same season.

The major weakness of this study is the missing reliable measurement of organic compounds of different volatilities. However, I find that the authors are well aware of the limitations, which are acknowledged appropriately and discussed extensively. Therefore I recommend the article to be published in Atmospheric Physics and Chemistry with minor revisions.

**General comments and questions:**

-how representative were the 3 discussed cases in light of the whole measurement period? Can you estimate which growth pathway was the most important at this site? What determines the prevailing growth mechanism, the sulfuric acid concentration?

-you have particle size distribution measurements from 1.9nm-528nm, yet you report only the growth rate between 10-20nm, why? It would be very interesting to see how the growth rate (and the primary growth mechanism, if you can get that information) changes with particle size

-can you estimate how accurately was the sulfuric acid concentration measured and how the uncertainties in the total sulfuric acid concentrations affect your conclusions

-based on the text and the showed results it seems that each of the considered days had a predominant growth-mechanism, however, the other compounds also had a minor, but distinct contribution. Therefore, I think it is wrong to say 'growth by organics alone' (as you do in the abstract) for April 19.  I think you need to change the abstract and heading in chapter 3 to say 'Growth primarily by organics' and maybe even add 'with a small contribution from sulfuric acid and ammonia'

-the mechanism on May 9: should it be ammonium sulfate or sulfuric acid and ammonia? Maybe also mention the contribution from amines which seems to be non-negligible

-you claim that on May 11 the mechanism was sulfuric acid/amines/organics, although you say in the text you cannot assess the relative importance of ammonia to amines based on TDCIMS, and also MABNAG predicts both in the particle phase. So why not sulfuric acid, ammonia, amines and organics (or just call it mixed as in the synthesis chapter).

-there is currently almost no discussion on how the results of this study compare to other recent field and laboratory measurements about nano-particle growth rates, compositions and proposed growth mechanisms. I suggest the authors could include a short chapter on that before the conclusions section to give an idea how widely representative their results are.

**Specific comments:**

-your abstract is too long: you could leave out rows 26-30, which is more like introductory material and is indeed repeated in the introduction chapter.

-the four pathways mentioned in the abstract (row 26-29) and introduction (p.3, row 5-9) seem to be missing the interaction between sulfuric acid and bases (which is mentioned elsewhere, though)

-you could leave out 'analysis of' from the title. It would make it more concise and put emphasis on the fact that you found several different pathways (just a suggestion, though)

-p. 8, row 5. Please use SI units

-p. 13, row 10 citation

-it should be mentioned in the abstract and table 3 what size range you considered

-this reviewer found it confusing that you report a range of GRs for each event. Before carefully reading the methods I did not understand where this range came from. Why not give a mean value and then list all the considered size ranges (10-15, 15-20 and 10-20) in table 3, so you also get a feeling of the variation.

---

## Referee Comment (RC2) · Anonymous Referee #1 · 19 Apr 2016

This article presents an important study of growth pathways of nano-particles, comparing field measurements with results from process modeling. It is well writing and falls within the scope of the journal. While both modeling uncertainties and lack of some key measurements do not allow for closure, uncertainties and limitations are fully acknowledged and discussed. This discussion of limitations is an important aspect of the article, identifying key unknowns holding back out current understanding of nano-particle growth and signaling ways forward for future work. I recommend this article for publication in ACP with minor revisions.

Scientific Comments

1. GR methods

[Figure]

Page 12 lines 1-5: There are many methods that can be used to calculate GR from measured size distributions. Why is this method used as opposed to e.g. continuous dDp/dt using the mode diameter or using the leading edge of the banana instead of the mode diameter? An evaluation of how the assumptions in this method chosen compare with the way MABNAG calculates GR would show whether any of the systematic discrepancy between measured and modeled GRS is due to method differences instead of missing species as implicitly assumed in later sections. Part of the consistent under prediction of GR from MABNAG compared to these measurements could be due to systematic differences from the method of GR calculation. MABNAG measures wet diam. Temperature of saturators in CPCs of the SMPSs could mean measuring dry diam. For the compositions observed do you have an estimate of how much this could be affecting the particle diameters and thus GRs?

2. GR uncertainties

Page 12 paragraph 2: This range of GRs given by the 3 diameter ranges is not a measure of uncertainty, but is presented almost like an uncertainty in the results and evaluation section. Table 3 would be more useful if it showed which GR comes from which diameter range. Is there any trend in how GR varies over the diameter range? A more continuous dDp/dt plotted as a function of Dp would give a clearer view of this and could be usefully compared with how the MABNAG GRs vary with size and an experimental view on whether the reduction of the Kelvin effect with size (line 15) is a significant contributor or not.

Page 15 line 5: 50% MABNAG GR uncertainty from ELVOC concentration uncertainty would lead to a maximum GR of 2.1nmph, which is still lower than the measured 3nm GR. It needs to be stated clearly that this uncertainty in the ELVOC concentration cannot alone account for the model-measurement discrepancy in this case.

Page 15 line 9 and lines 16-19 : If LVOC and SVOC contribute more to growth as the particle size increases we would expect the modeled GR to deviate more from

the measured result at larger sizes, especially for the case where organics dominate growth. Is this apparent in the data?

Page 15 line 26: What is the uncertainty on the cluster CIMS SA measurement and also for the ammonia and amine concentrations on page 16 line 35?

Page 18 lines 27-36: RH uncertainty on MABNAG GR would be more usefully included quantitatively in the discussion of GRs for the separate case. This and the ELVOC concentration uncertainty (put at 50% earlier in the manuscript), the oxalic acid factor 100 uncertainty and perhaps other sources of uncertainty could allow a fuller basis for comparison between model and measurements if they were included quantitatively in the results.

Page 12 lines 5-10: Hypothesis that double nucleation is mixing of nucleation event that occurred higher up + later event lower down seems plausible for May 9th, but May 11th both events show particles growing from the smallest sizes. The plots in fig1. are composite plots from multiple SMPSs – could the larger concentrations at larger sizes on May 11th 1st nucleation event be a compatibility issue between SMPSs (i.e. could the SMPS measuring at larger sizes be measuring with a higher efficiency – perhaps because of unaccounted for diffusion losses in the sampling lines?)

May 11th 2nd nucleation event: condensation/coagulation sink from 1st nucleation event will likely be affecting the GR making it appear smaller than it would be for a single event – some estimation of the size of this effect would be useful as it would further increase the difference between the measured GR and the MABNAG modeled GR

3. Other

Page 7 paragraph 4: Some discussion on the accuracy of MEGAN2.1 estimations of monoterpene emissions and concentrations and applicability for this site/study is necessary to give confidence in their use.

Page 9 paragraph 1: What was the basis for choosing 20 molecules as the initial particle size? How does this choice affect the modeled results?

Page 12 line 5: On what is the assertion that nucleation potentially occurred aloft based?

Page 16 line 18: GR for ammonium sulfate case shows better model-measurement agreement than for the growth by organics case. SA/amine/organic growth on p17 also shows better agreement. Comment on why this might be?

Page 17 line 16: The base simulation predicts GR that are way too low compared to measured GR. Therefore the relevance of the composition from this model seems tenuous. In general, different MABNAG simulations have better/worse agreement with the measured GR and also predict different compositions – more could be made of which compositions are more likely to be accurate based on this.

Page 17 line 18: This sentence should include the fact that there are significant additional unknown growth pathways (N, LVOC, SVOC) as well

Technical Comments

Page 9 line 21: SVOC doesn't seem to be defined here or earlier and should be

Page 7 lines 10-11: "estimated uncertainty in oxalic acid . . . is approximately a factor 100 lower" is unclear. Need a better way of saying the oxalic acid concentrations could be up to 100x larger than measured as done later on page 11

Page 13 line 16: Reference needed for sulfuric acid concentration of 2e-6 leading to 0.2nmph growth rate

---

## Referee Comment (RC3) · Anonymous Referee #4 · 10 May 2016

This is a very nice paper that combines measurements with modelling. It is based in a technically challenging frontier of aerosol science: determining and verifying the composition of the smallest nucleating particles. It is generally well-defined and clearly written. The authors acknowledge the limitations of their methods and describe the approximations and assumptions they have relied on.

My major concern in this paper relates to the way in which the findings are communicated to the reader. Rather than having a huge data dump, it would be better to both group and separate the scenarios/simulations more clearly. Some methods are suggested below:

- In Tables 4-6, for each group of three simulations, the first and last identifiers are constant and the middle one changes. Either change the order of the identifiers or the order in which the simulations are presented. I think it would be clearer to have all the MAL simulations first, divided into thirds by 1ox/10ox/100ox, each of which is further divided based on DMA_L and TMA_L. After that, a similar breakdown of MAL_LoVP, then OX, then OX_LoVP. This would give a logical progression, while still leaving the reference scenario in second place on the list. However, the authors may prefer to use another system based on what they feel is the most important characteristic to group simulations for important comparisons. Based on my reading, it seems that the characteristics change from case to case, and so the ordering may be less important from that perspective.

- Add vertical lines between the case identifiers and growth rates, and between growth rates and mole fractions.

- Colour code simulations which differ significantly from the base case, or which provide the best reproduction of observations; at the very least, those which are discussed in-depth in the text. Refer to the cases by colour in the figure caption, for ease of understanding.

- Add explanatory text to the captions of Tables 4-6. As a general rule, a caption should provide enough information that the item can be at least minimally understood without any other context.

Using T to represent Total amines measured at SGP is quite confusing in the text, because it usually means Temperature (as it does in Table 1, for example). Maybe use TAm?

Use the format A.B $\times$ 10$^C$ in e.g. Table 1. ($\times$ in LaTeX, Insert →Symbol →× in MS Word)

In Figure 1, please add units to the colour bar for panels (a)-(c) (and it would be better to label the colour bar as "log N", or to only label whole powers of ten).

In panel (f) of Figures 2-4, ELVOCs and organic acids seem to be the same colour. Is this intentional? The mole fractions are shown separately in Tables 4-6, so I assume they can be distinguished at all

I would like to see an explicit equation for "the same calculations as used for April 19". I'm more of a physicist than a chemist, and while I tinkered around with the numbers in some of the tables, I couldn't reproduce 12.5% by mole. Of course, there were quite a lot of scenarios in the tables, so it was hard to be sure exactly which numbers I was meant to be using...

The growth rates listed in Table 3 show a single number, whereas the text references three different numbers for each day. It would be better to see those numbers explicitly rather than be given a range.

Aside from these minor concerns, I found the paper interesting and feel that it makes an important contribution to its field. I would recommend that it be published subject to minor revisions.

---

## Author Comment (AC1) · 7 Jul 2016

*Review of Analysis of multiple new-particle growth pathways observed at the US DOE Southern Great Plains field site by Hodshire et al.*

*This article adds to the understanding of particle growth processes in the atmosphere. The article is well written and the topic fits to the scope of the journal. The article gives valuable, yet qualitative information about the growth pathways of nanoparticles, which is derived from both measurement and modelling. I find it especially important that the authors show that the major growth mechanism can vary even at the same site and the same season.*

*The major weakness of this study is the missing reliable measurement of organic compounds of different volatilities. However, I find that the authors are well aware of the limitations, which are acknowledged appropriately and discussed extensively. Therefore I recommend the article to be published in Atmospheric Physics and Chemistry with minor revisions.*

**General comments and questions:**

*-how representative were the 3 discussed cases in light of the whole measurement period? Can you estimate which growth pathway was the most important at this site? What determines the prevailing growth mechanism, the sulfuric acid concentration?*

During the campaign, 13 possible new-particle formation events were observed. We selected the 3 events that we did because all of the instruments available were working and the TDCIMS was measuring the sizes of the growing nucleation mode. As our growth hypotheses are based on the particle compositions obtained from the TDCIMS, it was important to have data from the smaller, growing particles.  For the 10 events that were not analyzed in the paper, the TDCIMS was measuring at a large (~40 nm) particle size for the event and thus the particle compositions of the growing particles was not captured for the most part. Also, some of these events were not captured by our other instruments due to instrument failure.  There were a few events in which the TDCIMS captured particles at the end of the growth event (e.g. when the growing particles reached ~40 nm) but at the larger particle sizes, there is more potential for contamination from the background accumulation-mode aerosols as the TDCIMS tends to take in significantly larger particles than the nominal 40 nm particles at these larger size cuts. With all of these considerations, we do not have the ability to estimate the more important growth pathway(s) for SGP during this campaign.

Regarding what determines the prevailing growth mechanism: We have sulfuric acid measurements for the entire campaign except during occasional periods of instrument failure. We do not currently have ELVOC estimates for the entire campaign, as we obtained the monoterpene emissions data from MEGAN from GEOS-Chem for only the three days

considered. Nor do we have any measurements or estimates of higher volatility organics (e.g. LVOCs or SVOCs) that can contribute to growth at larger particle sizes, as has been often brought up in the text. We do see that for the day that showed growth by primarily organics, the ELVOC estimated concentration at the beginning of the event is the highest of the three days and the sulfuric acid concentration at the beginning of the event is the lowest of the three days and that for the ammonium sulfate growth day, the sulfuric acid concentration is the highest out of the three days.  Both the organic acid and ELVOC estimated concentrations are higher for the ammonium sulfate day than for the growth by sulfuric-acid/bases/organics, and yet we see significantly more organics for the growth by sulfuric-acid/bases/organics day. It is difficult to draw any conclusions based on only three days as to what determines the prevailing growth mechanism.

*-you have particle size distribution measurements from 1.9nm-528nm, yet you report only the growth rate between 10-20nm, why? It would be very interesting to see how the growth rate (and the primary growth mechanism, if you can get that information) changes with particle size*

We selected the range 10-20 nm for a few different reasons. We do not see significant growth past ~20 nm for April 19, and we wish to remain consistent in our analysis across the three days. Our analysis of particle composition is somewhat constrained to this smaller size range: our hypotheses of growth mechanisms are based upon the TDCIMS data. During the campaign, the TDCIMS was set to measure at ~40 nm mode diameter when new-particle formation events were not ongoing, and set to measure smaller particle sizes (usually around ~20 nm) when the onset of a  new-particle formation event was  detected. The smaller size selection was chosen in order to determine what species were in the freshly growing particles. Unfortunately, not all of the events were detected in real-time, and several new-particle formation events occurred without concurrent TDCIMS measurements in the smaller size ranges--these days were not a part of this paper's analyses. Thus we cannot categorically state how the primarily growth mechanism(s) change with particle size, since we do not have TDCIMS data that tracks the growing particles beyond the ~20 nm range for the new-particle formation events.

We have added the following to the text: "We have calculated the observed growth rates between 10-20 nm for each day of our analysis from the SMPS data (Figure 1, a-c). This size range is used since we constrain our analysis of particle composition to the TDCIMS data. During the NPFS campaign, the TDCIMS was set to measure at ~40 nm mode diameter when new-particle formation events were not ongoing. Then, when the onset of a new-particle formation event was detected, the TDCIMS was set to measure smaller particle sizes, around 20 nm mode diameter, in order to determine what species were in the freshly growing particles. Thus, our growth rate calculations represent the size range that the TDCIMS measured in during the events of our analysis."

*-can you estimate how accurately was the sulfuric acid concentration measured and how the uncertainties in the total sulfuric acid concentrations affect your conclusions*

The uncertainty on the Cluster CIMS SA measurements is given in the SI of Chen et al. (2013). We have added the following to the discussion of the Cluster CIMS: "The detection of sulfuric acid in the CIMS has been quantified and calibrated, and the uncertainties for the concentrations of the monomers and dimers of sulfuric acid are estimated to be factors of 1.5 and 3, respectively (Chen et al., 2013)."

Regarding how the uncertainties of total sulfuric acid concentrations affect our conclusions: consider April 19 as an example. If we assume irreversible condensation (reasonable, given the particle sizes), an accommodation coefficient of 1, and a temperature of ~10 C, the reported sulfuric acid concentration of $2 \times 10^6$ molecules cm$^{-3}$ will lead to a growth rate of ~0.1 nm hr$^{-1}$ by sulfuric acid condensation alone. The concentration of sulfuric acid dimer tends to be at least 2-3 orders of magnitude lower than the concentrations of sulfuric acid monomer throughout the campaign, and often falls beneath the detection limit. Thus, we'll assume that any uncertainty in the dimer concentrations are negligible compared to uncertainties in the monomer concentrations. Under these assumptions, the sulfuric acid could be up to ~$3 \times 10^6$ molecules cm$^{-3}$, leading to a growth rate of ~0.12 nm hr$^{-1}$, a ~20% increase in growth from sulfuric acid alone. We have updated our growth rate calculations (please see our response to your final comment for more details on the growth rates) to include three growth rate methods; for April 19, these three methods yield a possible growth rate range of 1.6-7.7 nm hr$^{-1}$. So even at the low end of this range, 1.6 nm hr$^{-1}$, the contribution to growth from sulfuric acid goes from contributing 6.25% to 7.5% towards the total growth rate with a 50% increase in sulfuric acid contribution. This difference is too small when compared to our other uncertainties (including what the actual observed growth rate is) to account for any possible underpredictions in the MABNAG-predicted growth rates.

Similarly, the growth rate for May 9, the day that shows the most growth from sulfuric acid, only has about 0.8 nm hr$^{-1}$ of growth coming from condensation of sulfuric acid, assuming the reported concentration of ~$2 \times 10^7$ molecules cm$^{-3}$. A 50% increase in the sulfuric acid concentration to ~$3 \times 10^7$ molecules cm$^{-3}$ leads to a growth rate of ~1.2 nm hr$^{-1}$, a 50% increase in the growth rate from sulfuric acid. Thus, the uncertainties in sulfuric acid contribute to smaller uncertainties in growth rates than the other uncertainties discussed in the text.

*-based on the text and the showed results it seems that each of the considered days had a predominant growth-mechanism, however, the other compounds also had a minor, but distinct contribution. Therefore, I think it is wrong to say 'growth by organics alone' (as you do in the*

*abstract) for April 19. I think you need to change the abstract and heading in chapter 3 to say 'Growth primarily by organics' and maybe even add 'with a small contribution from sulfuric acid and ammonia'*

We have modified the abstract and heading for chapter 3 to reflect that we see growth primarily from organics.

*-the mechanism on May 9: should it be ammonium sulfate or sulfuric acid and ammonia? Maybe also mention the contribution from amines which seems to be non-negligible*

We have changed the heading to be Growth by primarily sulfuric acid and ammonia; we indicate in the text of this subsection that amines (and organics) appear to make a small but non-negligible contribution, as well: "A small, but non-trivial, amount of organics and amines are seen in the particle phase as well."

*-you claim that on May 11 the mechanism was sulfuric acid/amines/organics, although you say in the text you cannot assess the relative importance of ammonia to amines based on TDCIMS, and also MABNAG predicts both in the particle phase. So why not sulfuric acid, ammonia, amines and organics (or just call it mixed as in the synthesis chapter).*

We have changed the abstract and the text to reflect that we see a contribution from bases, instead of only amines.

*-there is currently almost no discussion on how the results of this study compare to other recent field and laboratory measurements about nanoparticle growth rates, compositions and proposed growth mechanisms. I suggest the authors could include a short chapter on that before the conclusions section to give an idea how widely representative their results are.*

We have added the following brief section:

4. The Southern Great Plains: Comparison to other campaigns

The New Particle Formation Study provided unique insights into new-particle formation events for the region during the spring of 2013, as both gas-phase and particle-phase measurements were taken concurrently in order to assess the species contribution to growth. We see that from three days of the campaign where all instruments were running, three different

dominant growth mechanisms are present, from growth by primarily organics to growth by primarily ammonium sulfate to a mixture of growth from organics, sulfuric acid, and bases.

Previous field campaigns have taken place to similarly assess the growth of new-particle formation events in the continental boundary layer. A review paper by Kulmala et al. (2004) and references therein considered over 100 field campaigns, both long-term and intensive, primarily at continental boundary layer sites. Growth rates were found to be mainly within the 1-20 nm hr$^{-1}$ range in the mid-latitudes, and our events are within this range. Furthermore, for campaigns in which growth rates and gas-phase sulfuric acid were measured, it was found that sulfuric acid tended to account for only 10-30% of the observed growth rates (Kulmala et al., 2004); although water and ammonia accounted for some of the remaining growth, organic compounds are thought to comprise the remaining growth. Studies within the past few years have reported growth from either primarily organics (e.g. Smith et al., 2008b; Kuang et al., 2010; Riipinen et al., 2011; Pierce et al., 2012) or inorganic components, primarily sulfate or ammonium sulfate (e.g. Bzdek et al., 2012).

On-line particle-composition measurements of sub-micron aerosols are a relatively new and still-evolving measurement technique. Smith et al. (2004) reported the first such measurements, using the TDCIMS to examine 6-20 nm particles. Another recently developed instrument is the Nano Aerosol Mass Spectrometer (NAMS) (Wang et al., 2006; Wang and Johnston et al., 2006; Pennington and Johnson, 2012), which reports quantitative elemental composition of nanoparticles in the 10-30 nm range. Of the recent studies that have used combined gas-phase measurements with particle-phase measurements (using either the TDCIMS, NAMS, or both) to determine dominant growth mechanisms (e.g. Smith et al., 2008b; Bzdek et al., 2012; Bzdek et al., 2014), this study is, to our knowledge, unique in reporting distinctly different dominant growth pathways for separate yet temporally closely spaced new-particle growth events. However, it is highly unlikely that SGP is truly unique in this regard; instead the findings of this paper point towards the value of investigating more field sites influenced by mixtures of anthropogenic and biogenic emission using similar combinations of gas-phase and particle-phase measurements.

***Specific comments:***

*-your abstract is too long: you could leave out rows 26-30, which is more like introductory
material and is indeed repeated in the introduction chapter.*

We feel that lines 26-30 briefly supply important motivation for this particular piece in work, in
other words, assessing the contribution of growth from these different pathways. As such, we
have chosen to leave these sentences in the abstract.

*-the four pathways mentioned in the abstract (row 26-29) and introduction (p.3, row 5-9) seem
to be missing the interaction between sulfuric acid and bases (which is mentioned elsewhere,
though)*

We have altered the discussion on sulfuric acid in abstract to read, "condensation of sulfuric acid
vapor (and associated bases when available)", and the discussion in the intro to read,
"Irreversible condensation of sulfuric acid vapor (produced through gas-phase oxidation of $SO_2$
by the hydroxyl radical) is known to be a major contributor to growth. The effective equilibrium
vapor pressure of sulfuric acid in the presence of tropospheric water vapor is negligible
compared to ambient sulfuric acid concentrations (Marti et al., 1997), and sulfuric acid readily
condenses to the smallest stable particles, often forming inorganic salts with associated bases
when available."

*-you could leave out 'analysis of' from the title. It would make it more concise and put emphasis
on the fact that you found several different pathways (just a suggestion, though)*

Done.

*-p. 8, row 5. Please use SI units*

Done.

*-p. 13, row 10 citation*

This was fixed after the pre-ACPD review.

*-it should be mentioned in the abstract and table 3 what size range you considered*

Done.

*-this reviewer found it confusing that you report a range of GRs for each event. Before carefully reading the methods I did not understand where this range came from. Why not give a mean value and then list all the considered size ranges (10-15, 15-20 and 10-20) in table 3, so you also get a feeling of the variation.*

We have reconsidered our GR methods for this work. We have completed the leading edge method and the mode diameter method for each day, as well as made a linear growth rate based upon visual inspection, all for the $D_p$ range or 10-20 nm. We have made figures showing the results of each method (see below) and have included these figures in the supplementary information. It can be seen that the leading edge and the $D_p$ mode methods, although fully automated and thus theoretically better than the visual method, do not always track the growing distribution well. We have added the following text to the discussion on calculating the observed growth rates:

"There is considerable noise in the SMPS data (Figure 1, a-c), especially for May 9 and May 11, due possibly to the hypothesized mixing down of particles and possible inhomogeneities in the air mass. For this reason, we have calculated the growth rate between 10-20 nm for each using three different methods. The first method, referred to here as the leading edge method, is adapted from Lehtipalo et al. (2014) and finds the time at which the binned aerosol distribution between 10-20 nm reaches one half of its maximum $dN/dlogD_p$ for each bin. A linear fit between the bin's median diameter and the associated time determines the growth rate. The second method, referred to here as the $D_p$-mode method, tracks the change in diameter of the maximum $dN/dlogD_p$ of the aerosol size distribution between 10-20 nm; a linear fit between the diameters and time determines the growth rate. When plotted against the size distribution (see supplement, Figures S1-S3), it is seen that the leading edge and $D_p$ mode method both do not always track the growing size distribution well. For this reason, we have included a third method, which we call the visual method, in which we have made a linear growth rate between 10-20 nm for each day based upon visual inspection of the size distribution (see supplement, Figure S1-S3), using Eq. (3):

$$GR_{obs} = dDp/dt \sim= \Delta Dp/\Delta t \qquad (3)$$

These three methods provides a range of growth rates (Table 3) for the particles between 10-20 nm; the specific results for each day will be discussed in section 3. We do not attempt to provide uncertainty estimates for each method, due to the overall noise in the data. Instead, we present the ranges of calculated growth rates as a possible range of the actual growth rates. May 9 and May 11 tend to have higher growth rates: this could be from the influence of the continued

mixing down from nucleation aloft and not actually representative of the growth rates of the particles forming near the surface."

Regarding the SMPS measurements: the measurements were made at ambient dew point. If the temperature inside the trailer were equal to the ambient temperature, then measurements would have been carried out at ambient relative humidity as well. As an approximation, we have assumed that particle sizes in the SMPSs were equal to particle sizes in ambient air (i.e., water was neither lost nor gained; if particles were wet in the ambient they were equally wet in the DMAs) because the temperatures in the trailer should be close to that of the ambient temperatures. We have modified the text to make this more clear, "For all systems, filtered ambient air was used for the DMA sheath air, without adjusting the water vapor partial pressure. Therefore, the relative humidity was close to ambient relative humidity, and particle water content was close to that in the atmosphere."

[Figure]

Figure S1. The results of the three growth rate calculations for April 19, 2013. The x-axis represents CDT time. The line at 15 nm $D_p$ is to guide the eye.

[Figure]

Figure S2. The results of the three growth rate calculations for May 9, 2013. The x-axis represents CDT time. The line at 15 nm $D_p$ is to guide the eye.

[Figure]

Figure S3. The results of the three growth rate calculations for May 11, 2013. The x-axis represents CDT time. The line at 15 nm $D_p$ is to guide the eye.

---

## Author Comment (AC2) · 7 Jul 2016

*This article presents an important study of growth pathways of nanoparticles, comparing field measurements with results from process modeling. It is well writing and falls within the scope of the journal. While both modeling uncertainties and lack of some key measurements do not allow for closure, uncertainties and limitations are fully acknowledged and discussed. This discussion of limitations is an important aspect of the article, identifying key unknowns holding back out current understanding of nanoparticle growth and signaling ways forward for future work. I recommend this article for publication in ACP with minor revisions.*

**Scientific Comments 1. GR methods**

*Page 12 lines 1-5: There are many methods that can be used to calculate GR from measured size distributions. Why is this method used as opposed to e.g. continuous dDp/dt using the mode diameter or using the leading edge of the banana instead of the mode diameter? An evaluation of how the assumptions in this method chosen compare with the way MABNAG calculates GR would show whether any of the systematic discrepancy between measured and modeled GRS is due to method differences instead of missing species as implicitly assumed in later sections. Part of the consistent under prediction of GR from MABNAG compared to these measurements could be due to systematic differences from the method of GR calculation. MABNAG measures wet diam. Temperature of saturators in CPCs of the SMPSs could mean measuring dry diam. For the compositions observed do you have an estimate of how much this could be affecting the particle diameters and thus GRs?*

We have reconsidered our GR methods for this work. We have completed the leading edge method and the mode diameter method for each day, as well as made a linear growth rate based upon visual inspection, all for the $D_p$ range of 10-20 nm. We have made figures showing the results of each method (see below) and have included these figures in the supplementary information. It can be seen from the figures that the leading edge and the $D_p$-mode methods, although fully automated and thus theoretically better than the visual method, do not perfectly

track the growing distribution well. We have added the following text to the discussion on calculating the observed growth rates:

"There is considerable noise in the SMPS data (Figure 1, a-c), especially for May 9 and May 11, due possibly to the hypothesized mixing down of particles and possible inhomogeneities in the air mass. For this reason, we have calculated the growth rate between 10-20 nm for each using three different methods. The first method, referred to here as the leading edge method, is adapted from Lehtipalo et al. (2014) and finds the time at which the binned aerosol distribution between 10-20 nm reaches one half of its maximum $dN/dlogD_p$ for each bin. A linear fit between the bin's median diameter and the associated time determines the growth rate. The second method, referred to here as the $D_p$-mode method, tracks the change in diameter of the maximum $dN/dlogD_p$ of the aerosol size distribution between 10-20 nm; a linear fit between the diameters and time determines the growth rate. When plotted against the size distribution (see supplement, Figures S1-S3), it is seen that the leading edge and $D_p$ mode method both do not always track the growing size distribution well. For this reason, we have included a third method, which we call the visual method, in which we have made a linear growth rate between 10-20 nm for each day based upon visual inspection of the size distribution (see supplement, Figure S1-S3), using Eq. (3):

$$GR_{obs} = dDp/dt \sim= \Delta Dp/\Delta t \qquad (3)$$

These three methods provides a range of growth rates (Table 3) for the particles between 10-20 nm; the specific results for each day will be discussed in section 3. We do not attempt to provide uncertainty estimates for each method, due to the overall noise in the data. Instead, we present the ranges of calculated growth rates as a possible range of the actual growth rates. May 9 and May 11 tend to have higher growth rates: this could be from the influence of the continued mixing down from nucleation aloft and not actually representative of the growth rates of the particles forming near the surface."

In re the SMPS measurements: the measurements were made at ambient dew point. If the temperature inside the trailer were equal to the ambient temperature, then measurements would have been carried out at ambient relative humidity as well. As an approximation, we have assumed that particle sizes in the SMPSs were equal to particle sizes in ambient air (i.e., water was neither lost nor gained; if particles were wet in the ambient they were equally wet in the DMAs). The temperatures in the trailer should be close to that of the ambient temperatures. We have modified the text to make this more clear, "For all systems, filtered ambient air was used for the DMA sheath air, without adjusting the water vapor partial pressure. Therefore, the

relative humidity was close to ambient relative humidity, and particle water content was close to that in the atmosphere."

[Figure]

Figure S1. The results of the three growth rate calculations for April 19, 2013. The x-axis represents CDT time. The line at 15 nm $D_p$ is to guide the eye.

[Figure]

Figure S2. The results of the three growth rate calculations for May 9, 2013. The x-axis represents CDT time. The line at 15 nm $D_p$ is to guide the eye.

[Figure]

Figure S3. The results of the three growth rate calculations for May 11, 2013. The x-axis represents CDT time. The line at 15 nm $D_p$ is to guide the eye.

**2. GR uncertainties**

*Page 12 paragraph 2: This range of GRs given by the 3 diameter ranges is not a measure of uncertainty, but is presented almost like an uncertainty in the results and evaluation section. Table 3 would be more useful if it showed which GR comes from which diameter range. Is there any trend in how GR varies over the diameter range? A more continuous dDp/dt plotted as a function of Dp would give a clearer view of this and could be usefully compared with how the MABNAG GRs vary with size and an experimental view on whether the reduction of the Kelvin effect with size (line 15) is a significant contributor or not.*

See our response to the first half of your first comment under **Scientific Comments 1. GR methods** in re the issue of using a range of growth rates; we instead have chosen to present a variety of growth rate methods and their results to show the possible variety in growth rates. Given the variety from just using different methods due to the noise in the SMPS data, we are hesitant to attempt to calculate growth rates at different size ranges, as this will introduce further uncertainties. We do not feel that we can confidently determine any changes in the growth rate from possible Kelvin effects at smaller sizes based on these uncertainties.

*Page 15 line 5: 50% MABNAG GR uncertainty from ELVOC concentration uncertainty would lead to a maximum GR of 2.1 nmph, which is still lower than the measured 3nm GR. It needs to be stated clearly that this uncertainty in the ELVOC concentration cannot alone account for the model-measurement discrepancy in this case.*

We have added the following to the discussion: "However, even a 50% underprediction of the contribution from ELVOCs to growth would lead to a maximum growth rate of 2.1 nm hr$^{-1}$, which is on the low end of the growth-rate range that we have calculated from the measurements. Thus, our low bias in growth rate for this day may not be from the ELVOC concentration uncertainties alone."

*Page 15 line 9 and lines 16-19 : If LVOC and SVOC contribute more to growth as the particle size increases we would expect the modeled GR to deviate more from the measured result at larger sizes, especially for the case where organics dominate growth. Is this apparent in the data?*

For the three days that are analysed here, getting growth rates much past 20 nm in diameter is tenuous, at best. By the time all three days begin to reach sizes beyond 20 nm in diameter, the amount of noise in the SMPS data increases, making the automated growth rate methods' results (e.g. the leading edge and $D_p$-mode methods) more uncertain. Furthermore, none of the days analysed here grow significantly past ~35 nm. April 19 does not grow far enough past 20 nm to confidently get a growth rate even from the visual method. (The strong growth seen until about hour 18 ends at around 22 nm $D_p$.)

Similarly, the strong growth seen for the 2nd event of May 9 ends at around 20 nm by about hour 15; the SMPS data at higher $D_p$ at later times shows a very similar number concentration across several $D_p$ bins, making it difficult to pick out a possible growth events.

It is possible to estimate the growth rate for May 11 using the visual method; we obtain a growth rate of around 3 nm hr$^{-1}$ between 20-30 nm between hours ~16-19. This is slower than the growth rate calculated for May 11 using the visual method between 10-20 nm; that growth rate was found to be 8.3 nm hr$^{-1}$. (It should be noted that for May 11, we likely have a lot of pre-existing aerosol from the first event in the 20-30 nm bins during this time period, making growth-rate calculations less trustworthy.) Across cases and days, if the aerosol grows past 20 nm, MABNAG predicts a growth rate in the 20-30 nm range that is essentially the same (usually slightly slower but only by a few percent) as what is predicted for the 10-20 nm range. Given our uncertainties in the SMPS data, we cannot estimate whether or not the modelled GRs deviate more from the measured GRs at larger sizes.

*Page 15 line 26: What is the uncertainty on the cluster CIMS SA measurement and also for the ammonia and amine concentrations on page 16 line 35?*

The uncertainty on the Cluster CIMS SA measurements is given in the SI of Chen et al. (2013). We have added the following to the discussion of the Cluster CIMS: "The detection of sulfuric acid in the CIMS has been quantified and calibrated, and the uncertainties for the concentrations of the monomers and dimers of sulfuric acid are estimated to be factors of 1.5 and 3, respectively (Chen et al., 2012)."

The uncertainty for the data set from the AmPMS for the SGP campaign (ammonia and amine concentrations) is given in the SI of Freshour et al. (2014). We have added the following to the discussion of the AmPMS: "Uncertainties in the AmPMS data for this campaign is discussed further in Freshour et al. (2014) and is estimated to be +150/-60%, overall."

*Page 18 lines 27-36: RH uncertainty on MABNAG GR would be more usefully included quantitatively in the discussion of GRs for the separate case. This and the ELVOC concentration uncertainty (put at 50% earlier in the manuscript), the oxalic acid factor 100 uncertainty and perhaps other sources of uncertainty could allow a fuller basis for comparison between model and measurements if they were included quantitatively in the results.*

One major difficulty in qualitatively including uncertainties for each day for the different uncertain variables is that we do not know the PDFs of these uncertainties. For instance, is the likelihood of oxalic acid equally distributed between 1x and 100x? Could it in reality be lower than 1x or higher than 100x--and so forth for the other uncertain factors. It is beyond the scope of this paper to attempt to fully map out each uncertain variable in its entirety, and we currently do not have a good enough understanding of the uncertainty spaces of our parameters. Instead, discussing the qualitative uncertainties addresses what limitations we have in this study, without introducing further layers of uncertainty by attempting to quantify each uncertain parameter.

*Page 12 lines 5-10: Hypothesis that double nucleation is mixing of nucleation event that occurred higher up + later event lower down seems plausible for May 9th, but May 11th both events show particles growing from the smallest sizes. The plots in fig1. are composite plots from multiple SMPSs – could the larger concentrations at larger sizes on May 11th 1st nucleation event be a compatibility issue between SMPSs (i.e. could the SMPS measuring at larger sizes be measuring with a higher efficiency – perhaps because of unaccounted for diffusion losses in the sampling lines?)*

We put a large effort into comparing the responses of the 3 SMPS instruments for the size ranges where they overlap, and it was seen that the SMPS systems are in reasonable agreement for these overlapping size ranges. Thus we are confident that the measurements are reasonably accurate, and that the higher concentrations at larger sizes for the first nucleation event of May 11 are not due to SMPS compatibility issues.

*May 11th 2nd nucleation event: condensation/coagulation sink from 1st nucleation event will likely be affecting the GR making it appear smaller than it would be for a single event – some estimation of the size of this effect would be useful as it would further increase the difference between the measured GR and the MABNAG modeled GR*

MABNAG does not rely on the condensation sink explicitly to make its predictions.  We use the measured gas-phase species to drive MABNAG.  The condensation sink from the first event would have affected the measured gas-phase sulfuric acid etc on that day, and thus this condensation sink is implicitly included in the MABNAG simulations.

**3. Other**

*Page 7 paragraph 4: Some discussion on the accuracy of MEGAN2.1 estimations of monoterpine emissions and concentrations and applicability for this site/study is necessary to give confidence in their use.*

Guenther et al. (2012) provides an discussion of the various uncertainties related to the MEGAN2.1 model; however, they defer to Lamb et al. (1987) and state that , "the uncertainty estimate of a factor of three can be associated with the annual global emissions" of monoterpenes. However, as far as we are aware, no regional studies have been done to determine what uncertainties should be associated with the SGP region in regards to MEGAN model output. Thus, an uncertainty factor of three for the annual global emissions is the best we have at the moment.

We have added the following to the methods discussion of MEGAN: "For a discussion on the uncertainties associated with emissions from MEGAN2.1, see Guenther et al. (2012). "

*Page 9 paragraph 1: What was the basis for choosing 20 molecules as the initial particle size? How does this choice affect the modeled results?*

Choosing 20 molecules of each species as the initial particle size is arbitrary and creates a particle that is ~3 nm in diameter. The choice of molecules in the 3 nm particle has negligible influence on the growth rate and composition in the 10-20 nm size range.

We have added the following to the text: "MABNAG also requires an initial particle size and composition; for simplicity in this study, the initial particle is formed from 20 molecules of each input species, creating a particle approximately 3 nm in diameter. The choice of molecules in forming the initial particle has negligible influence on the growth rate and composition in the 10-20 nm size range."

*Page 12 line 5: On what is the assertion that nucleation potentially occurred aloft based?*

The assertion that nucleation potentially occurred aloft is based on tethered balloon data that measured particle size distributions that was taken on May 12. We have added further details to the text as follows: "Similar to May 9 and May 11, the SMPS data for May 12 shows what appears to be two nucleation events occurring at the surface where the SMPS collected size distributions. Tethered-balloon flight profiles for May 12 indicate that nucleation potentially occurred aloft. These observations will be described in detail in a manuscript currently in preparation (Craig, et al., 2016, in preparation), but are briefly described here: The balloon payload consisted of two portable condensation particle counters (model 3007, TSI, Inc.) operating at different minimum size-cut points, which allowed the vertically-resolved measurement of 10 to 20 nm diameter particle number concentrations, $N_{10-20nm}$. On May 12, high concentrations of particles in this size range were detected at 600 m above ground level, exactly coincident with, or slightly prior to, ground-level observations of high concentrations of $N_{10-20nm}$. We hypothesize the following explanation for the "double" nucleation events observed on May 9, 11, and 12: Nucleation and growth begins to occur aloft in the residual layer. Once the mixed-layer depth grows into the residual layer, these new particles (that may have already grown to ~10 nm) then mix down and are measured at the surface. This hypothesis is supported by the presence of a high concentration of larger particles ($D_p$ = 10-30 nm) that have already undergone growth at the "beginning" of the first event as measured by the SMPS on May 9 and May 11. Then, the second event, which presumably begins near the surface, shows a high concentration of freshly growing particles (3-5 nm, close to the limit of the SMPS detection) before larger particles appear."

*Page 16 line 18: GR for ammonium sulfate case shows better model-measurement agreement than for the growth by organics case. SA/amine/organic growth on p17 also shows better agreement. Comment on why this might be?*

We expect to do well for an inorganic ammonium-sulfate system, as gas-phase measurements are complete, and the thermodynamic properties and chemical interactions of ammonia/ammonium and sulfuric acid/sulfate are well-known. Growth by organics is much more difficult to constrain especially under this particular modelling framework given the uncertainties discussed throughout the paper (properties, LVOCs/SVOCs etc.). Given these limitations, it is unsurprising that we do not model the growth rates well for the growth by organics case.

We do not necessarily agree that we do much better for the SA/base/organics growth case than we do for the growth by organics case. We appear to do worse in terms of underpredicting growth rates for the SA/base/organics growth case than we do for the organics case. However, we do see a reasonable amount of sulfate and base in the particle phase for many of the sensitivity cases, which once again is likely due to our ability to model inorganic ammonium-sulfate cases well.

*Page 17 line 16: The base simulation predicts GR that are way too low compared to measured GR. Therefore the relevance of the composition from this model seems tenuous. In general, different MABNAG simulations have better/worse agreement with the measured GR and also predict different compositions – more could be made of which compositions are more likely to be accurate based on this.*

We do not see one set of assumptions in MABNAG that best captures all three days, and the only cases that had completely unreasonable results were the MAL_LoVP/100ox cases for May 9. Many cases tended to predict similar particle compositions and growth rates with only slight differences from case to case. Given this, the qualitative nature of the TDCIMS data, and that we are not including higher volatility organics (LVOCs and SVOCs) in the model and do not know whether we are accounting for the nitrogen species whose signal often shows up strongly in the TDCIMS, we cannot make definite model-to-observational comparisons and instead present these results as the basis of further research, especially into the areas that we are limited by (i. e. higher volatility organics, etc.).

We have added the following to the discussion for May 11:
"We do note that as MABNAG appears to be underpredicting the growth rates more than for April 19 or May 9 that the MABNAG-predicted particle compositions (Figures 6 and 7) are possibly less representative of the actual particle compositions. However, we reiterate our hypothesis that the underpredictions could be from the nitrogen-containing species that are detected in the TDCIMS but are not accounted for in MABNAG, as well as our uncertainty in ELVOC concentrations and lack of LVOCs, SVOCs, and accretion reactions. Furthermore, this day shows a more variable particle-phase spectrum than April 19 or May 9, as well as a more poorly defined second growth event (Figure 1c), making the observed growth rates difficult to

determine. The TDCIMS particle composition information is only qualitative. Thus, we will not speculate what differences are possible between observed and modelled particle composition."

We have also added the following sentence to the end of the first paragraph of our synthesis section: "We do not see that one set of assumptions in MABNAG best captures all three days (Figures 3, 5, and 7), and instead present these results as a basis for further research, especially into the contribution of higher-volatility organic species to growth."

*Page 17 line 18: This sentence should include the fact that there are significant additional unknown growth pathways (N, LVOC, SVOC) as well*

We do mention the N uncertainty in the previous paragraph, "The TDCIMS negative ion data also indicate the presence of nitrate; as stated previously, we hesitate to attribute significant growth from nitrate due to the unknown sensitivity of the TDCIMS to nitrate." We have modified the paragraph that page 17/line 18 was in to read, " Conversely, MABNAG predicts roughly 5-25% of the moles in the particle to be from ELVOCs, with the lowest relative ELVOC contribution seen in MAL_LoVP/100ox cases. Since the TDCIMS shows a variable amount of organics throughout the event, and we do not know the actual individual contributions from ELVOCs and organic acids, nor are we accounting for any higher-volatility neutral organic species (e.g. LVOCs and SVOCs), we cannot conclude which set of organics inputs best captures this day and do not exclude any set of inputs for being unrealistic."

**Technical Comments**

*Page 9 line 21: SVOC doesn't seem to be defined here or earlier and should be*

Done

*Page 7 lines 10-11: "estimated uncertainty in oxalic acid . . . is approximately a factor 100 lower" is unclear. Need a better way of saying the oxalic acid concentrations could be up to 100x larger than measured as done later on page 11*

We have modified the text: "Therefore, the estimated systematic uncertainty in the oxalic acid concentration measured via nitrate chemical ionization is approximately up to a factor 100 times lower than reported, indicating that the actual concentration could be up to 100 times higher than observed."

*Page 13 line 16: Reference needed for sulfuric acid concentration of 2e-6 leading to 0.2nmph growth rate*

This GR was calculated using the growth rate formula for the kinetic regime,

$$dDp/dt = ([H_2SO_4])*(Mw*c*alpha) / (2*rho)$$

Assuming accommodation coefficient alpha = 1; c = sqrt( (R*T)/(pi*Mw)); assuming T = 283 K; R = gas constant; Mw = molecular weight of sulfuric acid; $[H_2SO_4]$ = concentration of sulfuric acid; and rho = density of sulfuric acid. This leads to a GR closer to about 0.1 nm hr$^{-1}$; we have altered the text to reflect that as well as give more information.

The text now reads as follows: "Some notable features of the gas-phase data for April 19 (Figure 2a-b) include relatively low sulfuric-acid concentrations ($\sim$2$\times$10$^6$ cm$^{-3}$), which should only contribute to growth rates of about 0.08 nm hr$^{-1}$ (assuming kinetic regime growth, an accommodation coefficient of 1, and a temperature of 283 K), or approximately 10% of the observed rates."

---

## Author Comment (AC3) · 7 Jul 2016

*This is a very nice paper that combines measurements with modelling. It is based in a technically challenging frontier of aerosol science: determining and verifying the composition of the smallest nucleating particles. It is generally well-defined and clearly written. The authors acknowledge the limitations of their methods and describe the approximations and assumptions they have relied on.*

*My major concern in this paper relates to the way in which the findings are communicated to the reader. Rather than having a huge data dump, it would be better to both group and separate the scenarios/simulations more clearly. Some methods are suggested below:*

*In Tables 4-6, for each group of three simulations, the first and last identifiers are constant and the middle one changes. Either change the order of the identifiers or the order in which the simulations are presented. I think it would be clearer to have all the MAL simulations first, divided into thirds by 1ox/10ox/100ox, each of which is further divided based on DMA_L and TMA_L. After that, a similar breakdown of MAL_LoVP, then OX, then OX_LoVP. This would give a logical progression, while still leaving the reference scenario in second place on the list. However, the authors may prefer to use another system based on what they feel is the most important characteristic to group simulations for important comparisons. Based on my reading, it seems that the characteristics change from case to case, and so the ordering may be less important from that perspective.*

We have changed Tables 4-6 to Figures 3, 5, and 7. Figures 3, 5, and 7 display both the mole fractions and mass fractions of each species' contributions to the particle for each case. This more clearly communicates how the relative amounts of each species may or may not change across different assumptions.  We have further grouped all malonic cases together and all oxalic cases together, instead of grouping MAL/OX then MAL_LoVP/OX_LoVP cases together.

*Add vertical lines between the case identifiers and growth rates, and between growth rates and mole fractions.*

ACP provides strict formatting guidelines for tables; unfortunately, "Vertical lines must be avoided". However, we have translated these tables into Figures 3, 5, and 7, thus removing the tables entirely.

*Please colour code the simulations which differ significantly from the base case, or which provide the best reproduction of observations; at the very least, those which are discussed in-depth in the text. Refer to the cases by colour in the figure caption, for ease of understanding.*

Deviations should hopefully be more apparent with the new figures (Figures 3, 5, and 7).

*Add explanatory text to the captions of Tables 4-6. As a general rule, a caption should provide enough information that the item can be at least minimally understood without any other context.*

Tables 4-6 have been replaced with Figures 3, 5, and 7. We believe that the captions provided for each of these figure gives a clear explanation of each case and case identifier.

*Using T to represent Total amines measured at SGP is quite confusing in the text, because it usually means Temperature (as it does in Table 1, for example). Maybe use TAm?*

We agree and have changed all instances of T and L to Tam and Lam in the text and figures.

*Use the format A.B × 10C in e.g. Table 1. ($\times$ in LaTeX, Insert →Symbol →× in MS Word)*

Your suggested format is correct under ACP guidelines. The numbers have been reformatted in both the tables and text.

*In Figure 1, please add units to the colour bar for panels (a)-(c) (and it would be better to label the colour bar as "log N", or to only label whole powers of ten).*

Done.

*In panel (f) of Figures 2-4, ELVOCs and organic acids seem to be the same colour. Is this intentional? The mole fractions are shown separately in Tables 4-6, so I assume they can be distinguished at all.*

The ELVOCs are a dark green color, and the organic acids are a bright lime green color. There is so little organic acid in the particle phase throughout the shown MABNAG simulations that it cannot be seen for Figure 2 or 6; the predicted organic acid mole fraction is slightly distinguishable in Figure 4.

*I would like to see an explicit equation for "the same calculations as used for April 19". I'm more of a physicist than a chemist, and while I tinkered around with the numbers in some of the tables, I couldn't reproduce 12.5% by mole. Of course, there were quite a lot of scenarios in the tables, so it was hard to be sure exactly which numbers I was meant to be using...*

We have determined that our calculations for the formation of organic salts are tenuous, given the many uncertainties associated with the organic acids as model inputs  (e.g. concentration and chemical properties uncertainties). It is clear that (excluding the few unrealistic cases in which organic acid dominated the particle growth and particle growth rates exceeded 40-50 nm hr$^{-1}$) organic acids tend to contribute very little to the particle on both a molar and mass basis (see Figures 3, 5, and 7). Thus, we have removed the detailed discussions upon the possible numerical upper bounds of organic salt contribution to particle growth for each day and instead have made note of the small contributions of organic acids to particle growth and thus small contributions of organic salts to particle growth for each case day. For April 19, we state, "The majority of our simulations predict that less than 1% of the particle is organic acid by mole; thus, the contribution to particle growth from organic salt formation would be negligible, even when including the contribution from associated bases. Thus, we expect the majority of growth from organics to be coming from non-reactive organics (ELVOCs in our simulations) for this day." For May 9, we state, "However, given that most cases predict negligible (<3% by mole) of the particle to be composed of organic acid, the contribution to particle growth from organic salt formation is still predicted to be minor for this day." And for May 11, we state, "The majority of our simulations predict <5% by mole of the particle to be organic acid, thus again leading to only minor contributions from organic salt formation to particle growth."

*The growth rates listed in Table 3 show a single number, whereas the text references three different numbers for each day. It would be better to see those numbers explicitly rather than be given a range.*

We have revised our growth rate estimates for this work. We have decided to use three different methods of determining the growth rate: the leading edge method, the Dp mode method, and a visual method. We have inserted the following into the discussion on calculating observed growth rates:

"There is considerable noise in the SMPS data (Figure 1, a-c), especially for May 9 and May 11, due possibly to the hypothesized mixing down of particles and possible inhomogeneities in the air mass. For this reason, we have calculated the growth rate between 10-20 nm for each using three different methods. The first method, referred to here as the leading edge method, is adapted from Lehtipalo et al. (2014) and finds the time at which the binned aerosol distribution between 10-20 nm reaches one half of its maximum $dN/dlogD_p$ for each bin. A linear fit between the bin's median diameter and the associated time determines the growth rate. The second method, referred to here as the $D_p$-mode method, tracks the change in diameter of the maximum $dN/dlogD_p$ of the aerosol size distribution between 10-20 nm; a linear fit between the diameters and time determines the growth rate. When plotted against the size distribution (see supplement, Figures S1-S3), it is seen that the leading edge and $D_p$ mode method both do not always track the growing size distribution well. For this reason, we have included a third method, which we call the visual method, in which we have made a linear growth rate between 10-20 nm for each day based upon visual inspection of the size distribution (see supplement, Figure S1-S3),  using Eq. (3):

$$GR_{obs} = dDp/dt \sim= \Delta Dp/\Delta t \qquad (3)$$

These three methods provides a range of growth rates (Table 3) for the particles between 10-20 nm; the specific results for each day will be discussed in section 3. We do not attempt to provide uncertainty estimates for each method, due to the overall noise in the data. Instead, we present the ranges of calculated growth rates as a possible range of the actual growth rates.  May 9 and May 11 tend to have higher growth rates: this could be from the influence of the continued mixing down from nucleation aloft and not actually representative of the growth rates of the particles forming near the surface."

We have also included in the supplement a figure for each day that shows the results and best-fit lines of these three methods, included below.

[Figure]

Figure S1. The results of the three growth rate calculations for April 19, 2013. The x-axis represents CDT time. The line at 15 nm $D_p$ is to guide the eye.

[Figure]

Figure S2. The results of the three growth rate calculations for May 9, 2013. The x-axis represents CDT time. The line at 15 nm $D_p$ is to guide the eye.

[Figure]

Figure S3. The results of the three growth rate calculations for May 11, 2013. The x-axis represents CDT time. The line at 15 nm D$_p$ is to guide the eye.

*Aside from these minor concerns, I found the paper interesting and feel that it makes an important contribution to its field. I would recommend that it be published subject to minor revisions.*